# Circadian clock mechanism driving mammalian photoperiodism

S. H. Wood [1,2,8], M. M. Hindle [3,8], Y. Mizoro[1], Y. Cheng [4,5], B. R. C. Saer[1], K. Miedzinska[3], H. C. Christian [6], N. Begley[1], J. McNeilly[7], A. S. McNeilly[7], S. L. Meddle [3], D. W. Burt [3,4] & A. S. I. Loudon [1✉]

The annual photoperiod cycle provides the critical environmental cue synchronizing rhythms of life in seasonal habitats. In 1936, Bünning proposed a circadian-based coincidence timer for photoperiodic synchronization in plants. Formal studies support the universality of this so-called coincidence timer, but we lack understanding of the mechanisms involved. Here we show in mammals that long photoperiods induce the circadian transcription factor *BMAL2*, in the *pars tuberalis* of the pituitary, and triggers summer biology through the eyes absent/thyrotrophin (EYA3/TSH) pathway. Conversely, long-duration melatonin signals on short photoperiods induce circadian repressors including *DEC1*, suppressing BMAL2 and the EYA3/TSH pathway, triggering winter biology. These actions are associated with progressive genome-wide changes in chromatin state, elaborating the effect of the circadian coincidence timer. Hence, circadian clock-pituitary epigenetic pathway interactions form the basis of the mammalian coincidence timer mechanism. Our results constitute a blueprint for circadian-based seasonal timekeeping in vertebrates.

[1] Centre for Biological Timing, Faculty of Life Sciences, University of Manchester, Manchester M13 9PT, UK. [2] Arctic Chronobiology and Physiology Research Group, Department of Arctic and Marine Biology, UiT – The Arctic University of Norway, Tromsø 9037, Norway. [3] The Roslin Institute, and Royal (Dick) School of Veterinary Studies University of Edinburgh, Roslin, Midlothian EH25 9PRG, UK. [4] UQ Genomics Initiative, The University of Queensland, Brisbane, QLD 4072, Australia. [5] School of Life and Environmental Sciences, Faculty of Science, The University of Sydney, Camperdown, NSW, Australia. [6] University of Oxford, Department of Physiology, Anatomy and Genetics, Le Gros Clark Building, South Parks Road, Oxford OX1 3QX, UK. [7] MRC Centre for Reproductive Health, Queen's Medical Research Institute, Edinburgh EH16 4TJ, UK. [8] These authors contributed equally: S. H. Wood, M. M. Hindle. ✉email: andrew.loudon@manchester.ac.uk

The annual photoperiod cycle provides a critical predictive environmental cue driving annual cycles of fertility, physiology and behavior in most animal species. In 1936, Erwin Bünning proposed the external coincidence hypothesis for a circadian-based timing mechanism driving photoperiodic responses in plants[1]. The proposition was that photoperiod entrains a circadian rhythm of photosensitivity, and the expression of summer or winter biology depends on whether or not light coincides with the phase of high photosensitivity. An internal coincidence model has also been proposed where the role of light is to entrain two circadian oscillators and the phase relationship between the two oscillators determines the response to photoperiod[2,3]. In either case the role of the circadian clock is central and formal studies support the universality of a coincidence timer in animals[2–7], but we lack understanding of the mechanisms involved.

In mammals, the duration of the night-time pineal melatonin signal is sculpted by the photoperiod and is a critical regulator of annual cycles of reproduction, growth and metabolism[8–14]. The melatonin signal is decoded within the *pars tuberalis* (PT) of the pituitary gland[9,13,15–18]. Day-length dependent changes in hypothalamic thyroid hormone (TH) metabolism control the seasonal changes in physiology and are regulated via the PT, through altered secretion of thyrrotrophin (TSH)[19–21]. This TSH circuit is specific to the PT, and distinct from that in the anterior pituitary, which controls normal thyroid gland function. LP-activation of PT-TSH depends on regulation of TSHβ subunit expression by the transcriptional co-activator, EYA3, which operates as a TSH on-switch. EYA3 co-activates the PAR bZIP Transcription Factor TEF (thyrotroph embryonic factor) via a D-box element on the *TSHβ* promoter[22–24] (Fig. 1a). This prior work places EYA3 at the center of photoperiodic time measurement within the melatonin target tissue, but does not explain how expression is regulated by the seasonal clockwork.

In addition to acute circadian-based mechanisms an epigenetic basis to longer-term seasonal control in flowering in plants is well established. Specifically, the duration of cold temperatures during the winter season alters chromatin accessibility at key genes, and is a requirement for full photoperiod induced flowering (vernalization)[25–27]. In animals, no evidence for dynamic genome-wide seasonal epigenetic regulation of transcription has been described.

Here, we show that coincidence timing depends on a flip-flop switch between the expression of the circadian genes *BMAL2*, (a paralogue of the circadian regulator *BMAL1*) and *DEC1*, and their respective activating or suppressive effects on *EYA3* in the PT. The duration of the melatonin signal is key in sculpting these molecular components. Additionally, the effect of the coincidence timer is elaborated by progressive genome-wide changes in epigenetic status at key seasonal gene promoters. Therefore, circadian clock interactions with pituitary epigenetic pathways form the basis of the mammalian coincidence timer mechanism.

## Results

**Epigenetic regulation of the seasonal transcriptome.** Seasonally breeding sheep are a well-established photoperiodic model for the study of neuroendocrine mechanisms underpinning seasonal physiology[14,28,29]. Using a study design that compared the effects of transfer from long photoperiod (LP) to short photoperiod (SP) with transfer from SP to LP (Fig. 1b), we collected *pars tuberalis* (PT) tissue at 1,7 and 28 days after transfer, with collections timed for 4-h after lights on (ZT4), when *EYA3* expression peaks under LP[22,24,29]. Changes in prolactin concentrations[29–31] confirmed photoperiodic hormone responses (Fig. 1b, n = 28), as well as LP activation of *EYA3*, and inverse expression patterns of *TSHβ* (LP marker) and *CHGA* (SP marker) (Supplementary Fig. 1a–d, n = 4), validating this paradigm.

Comparing LP day 28 to SP day 28 by electron microscopy we found an increased nuclear diameter, equating to an approximate doubling in volume of PT thyrotrophs, and a marked reduction in chromatin density on LP (Fig. 1c, Supplementary Fig. 1e, f). These morphological changes were not seen in *Pars distalis* (PD) somatotrophs but were observed in the PT follicular stellate (FS) cells to a lesser degree (Fig. 1c, Supplementary Fig. 1e, f). We hypothesized that increased chromatin accessibility may account for the photoinductive effects of LP and therefore changes in the seasonal transcriptome.

Comparing all seasonal time-points (Fig. 1b, red arrows) we performed ChIP-seq (histone marker H3K4me3, Supplementary Data 1 and 2, n = 2) and RNA-seq (Supplementary Data 3, n = 3) to screen for seasonal transcriptional activation. To determine if epigenetic changes in H3K4me3 marks were associated with transcriptional activation we first improved the sheep genome annotation of transcripts and transcription start sites (TSSs) with PT RNA and a combination of Cap Analysis of Gene Expression (CAGE-seq, Supplementary Data 4)[32] and ISOSEQ long-read RNA-seq (SRA: PRJNA391103, see methods for details). Using this improved annotation we identified H3K4me3 peaks and found the distribution on genomic features to be similar to previous studies[33–35], validating our approach (see methods section).

Next we identified seasonally expressed genes, as defined by RNA-seq analysis of differentially regulated genes (DEGs) in the SP to LP and LP to SP transfers (Supplementary Data 3), and observed a strong correlation between seasonal gene expression and H3K4me3 peaks around the transcription start sites (TSSs) (Fig. 1d). Importantly, this correlation was absent in non-seasonally regulated genes and we found the distributions between seasonal and non-seasonally regulated genes to be significantly different (Fig. 1d, p-value <0.001 Mann–Whitney U-test). Histone modifications are precisely balanced by methyltransferases (writers), demethylases (erasers) and effector proteins (readers), therefore we checked the RNA expression of H3K4me3 readers, writers and erasers but found no seasonal changes (Supplementary Data 5). This suggests that changes in protein activity of H3K4me3 modulators may be key in the observed seasonal alterations of H3K4me3 marks.

We noted that approximately 70% of seasonally DEGs (Supplementary Data 3) had more than one TSS compared to only ~20% of the PT genomic background (Supplementary Fig. 1g, Supplementary Data 4). Next we took genes that were up-regulated in either SP or LP and plotted the proportion of genes with multiple TSSs, and repeated this for H3K4me3 marked TSSs, this revealed that H3K4me3 marks are more likely to occur on genes with multiple TSSs and highly expressed seasonal DEGs have a greater prevalence of multiple TSSs than non-seasonal genes expressed at the same level (Fig. 1e). Furthermore, this phenomenon was more pronounced in LP than SP. This indicates that LP up-regulated genes are more likely to have multiple TSSs and be associated with the H3K4me3 mark.

To investigate which promoter motifs may be actively transcribed we searched for enriched transcription factor binding site motifs in the TSSs marked with H3K4me3 in the LP day 28 vs SP day 28 comparison (Fig. 1f, Supplementary Fig. 2a, b). This showed enrichment of D-box binding motifs, Basic Leucine Zipper ATF-Like transcription related factors and POU domains for in both the up and down-regulated genes (Fig. 1f).

*TSHβ*, an exemplar seasonal D-box regulated gene[22,23], revealed progressive activation by LP, correlated with expression, and with no detectable H3K4me3 mark at either SP day 28 or 84 (R = 0.961, p-value = 0.002, Fig. 2a, b). In contrast, CHGA revealed an inverse seasonal pattern (R = 0.843, p-value = 0.009, Supplementary Fig. 2c, d). Global enrichment of D-box sites on

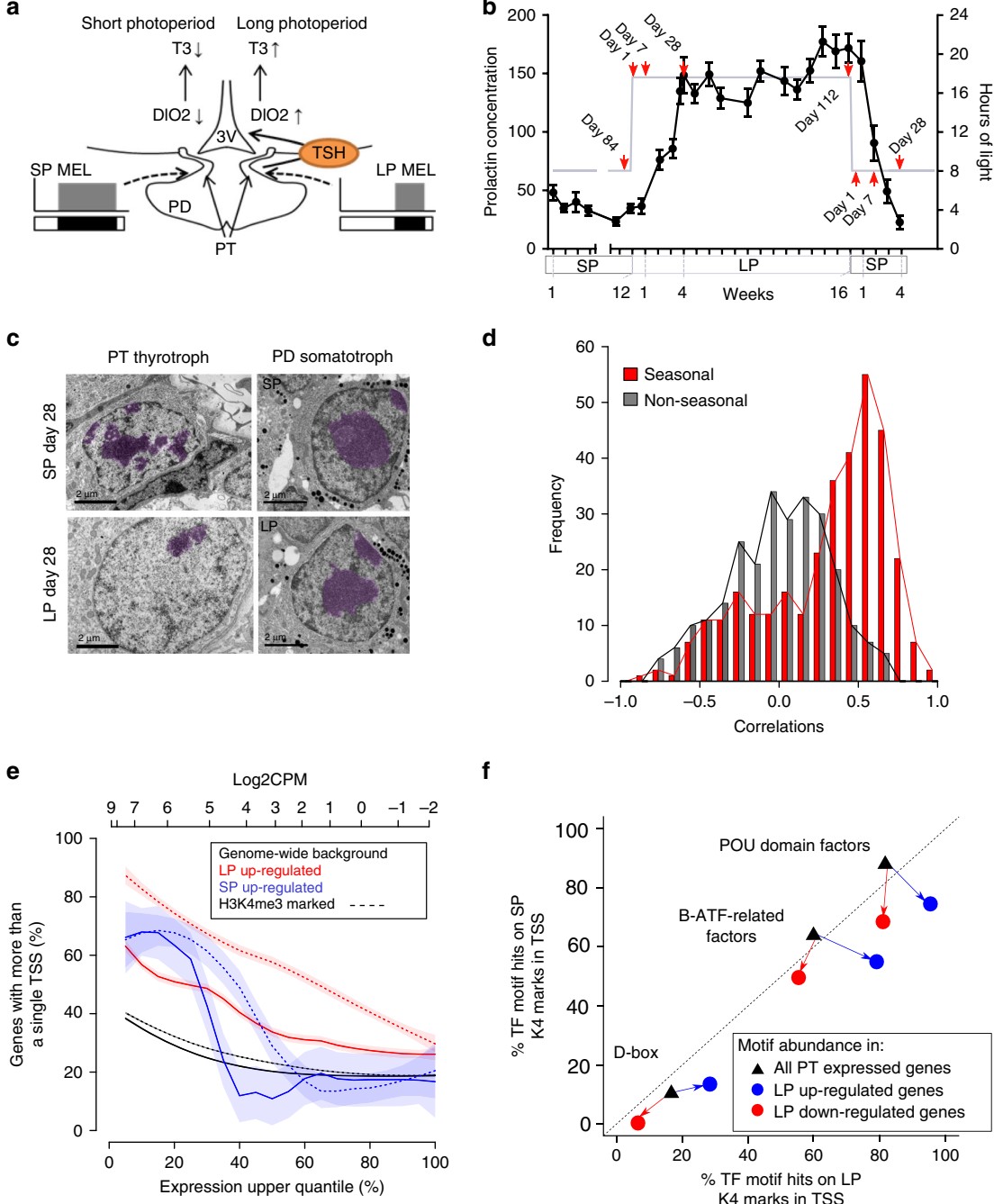

**Fig. 1 Photoperiod dependent epigenetic regulation of transcription in the *Pars tuberalis*. a** Current model for the mammalian photoperiodic circuitry. **b** Prolactin concentrations during the experiment (black), $n = 28$ biologically independent animals per time-point. Error bars represent the SEM. Gray line: number of hours of light the animals received in 24 hours. Red arrows show terminal sampling points (ZT4), with the exception of day 28 where animals were collected across the day at 4 hour intervals. **c** EM images of *Pars tuberalis* (PT) thyrotrophs and *Pars distalis* (PD) somatotrophs at LP day 28 and SP day 28. Dense chromatin in the nucleus is false colored in purple. Black scale bars $= 2 \mu m$. **d** Histogram revealing frequency distributions of Pearson correlation coefficients between RNA expression ($\log_2$ CPM) and H3K4me3 peak read counts $\pm 200$ bp from TSSs ($\log_2$ read counts). Red bars are seasonally expressed genes ($\log_2$ fold change $\geq 1$ or $\leq -1$ and adjusted *p*-value $<0.05$ of short photoperiod (SP) day 84 vs long photoperiod (LP) day 1, 7, 28 and LP day 112 vs SP day 1, 7, 28 differentially expressed genes (DEGs) = 480) and black bars are non-seasonally expressed genes ($\log_2$ fold change $\leq 0.1$ and $\geq -0.1$ from the same comparisons, (genes = 218)). **e** Prevalence of multiple (>1) TSSs in LP day 28 up-regulated DEGs (solid red), SP day 28 up-regulated DEGs (solid blue) and all PT expressed genes as the background (solid black) as a percentage of the uppermost expressed genes (i.e., increasing thresholds for the upper quantile of gene expression). $\log_2$CPM values for upper quantiles on upper x-axis. Dashed lines indicate the proportion of gene in the cohorts with multiple H3K4me3 (>1) marked TSS. **f** Over-represented motifs LP day 28 up and down-regulated genes, compared to SP day 28. The percentage of H3K4me3 marked TSS ($\pm 100$ bp) containing a specific motif is given. The black triangles represents the motif abundance in all the genes expressed (>0 CPM) in the PT and the percentage of marked with H3K4me3 in LP (*x*-axis) or SP (*y*-axis). The circles show the motif abundance and percentage H3K4me3 marked TSSs in LP up-regulated (blue) and down-regulated (red) genes (FDR <0.05; fishers two-way exact test).

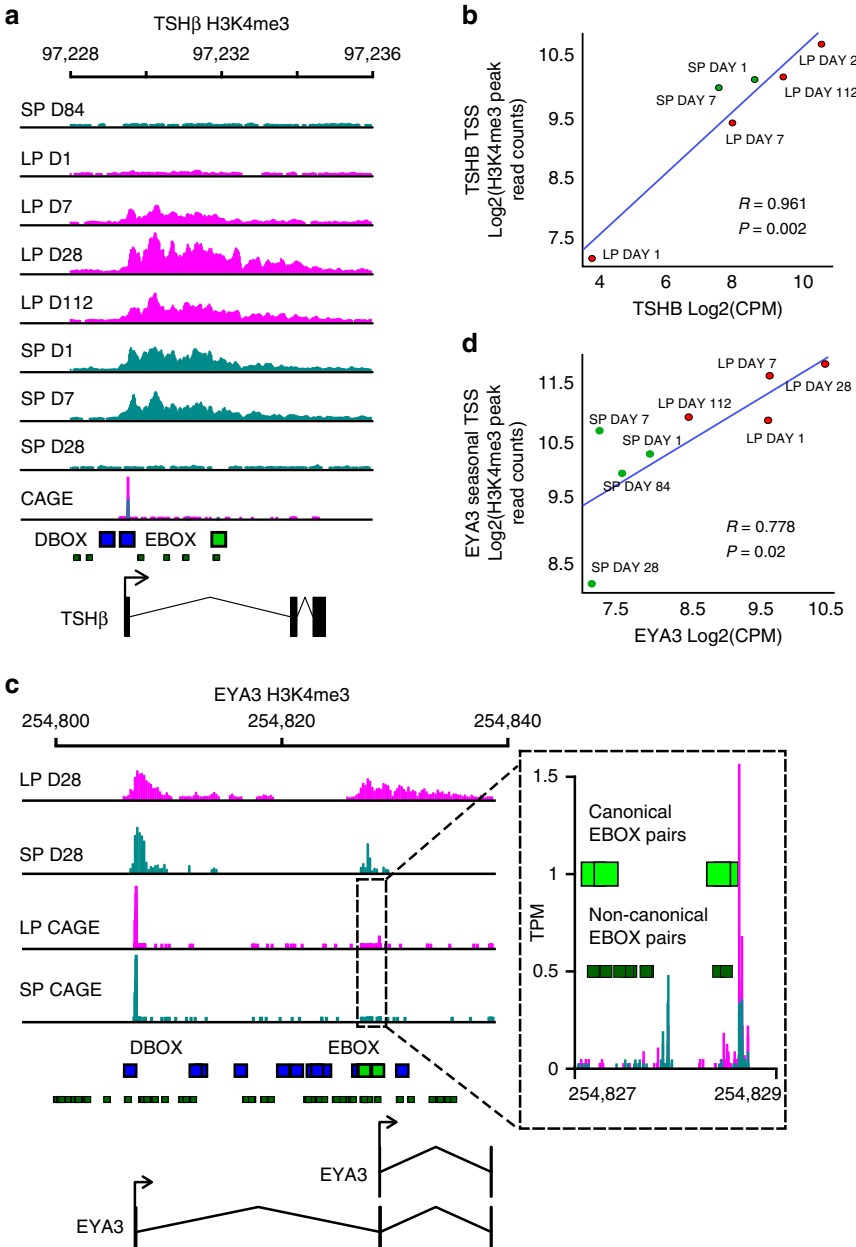

**Fig. 2 TSHβ and EYA3 are epigenetically regulated over the season. a** ChIP-seq tracks for *TSHβ* gene H3K4me3 peaks across all experimental time-points. Chromosome 1 region is shown. CAGE-seq track identifying active TSSs in LP day 28 vs SP day 28. Pink represents samples in long photoperiod and marine green represents short photoperiod. Solid green boxes are canonical E-box motifs. Blue boxes are D-box motifs. **b** Correlation plot for *TSHβ* log$_2$ H3k4me3 peaks from ChIP-seq versus *TSHb* log$_2$ counts per million (CPM) from RNA-seq. Red symbols are LP sampling points, green are SP sampling points. Correlation coefficient *R* is shown. $R = 0.961$, *p*-value $= 0.002$. Note: SP day 28 and 84 are not included because the H3K4me3 peaks are very low. **c** ChIP-seq tracks for *EYA3* gene H3K4me3 peaks. Chromsome 2 region is shown. CAGE-seq track identifying active TSSs in LP day 28 vs SP day 28. Zoom in box provided to identify the downstream promoter. **d** Correlation plot for *EYA3* downstream TSS log$_2$ H3K4me3 peaks from ChIP-seq versus *EYA3* log$_2$ CPM from RNA-seq. Red symbols are LP sampling points, green are SP sampling points. Correlation coefficient *R* is shown. $R = 0.778$, *p*-value $= 0.02$.

LP-activated genes indicates a potentially extensive role for TEF, SIX1 and EYA3 co-activation in seasonal regulation of physiology, and strongly focuses attention on the regulation of EYA3.

In line with our global analysis showing the presence of multiple TSSs in seasonally expressed genes we identified two transcription start sites in *EYA3*, an up-stream TSS with non-canonical E-boxes and a downstream TSS containing two paired canonical E-boxes[22] (Fig. 2c). CAGE analysis revealed that only the down-stream TSS was actively transcribed on LP (Fig. 2c,

Supplementary Data 4). We correlated RNA expression with H3K4me3 marks for both TSS, which confirmed seasonal regulation specific to the down-stream TSS (EYA3 downstream TSS: $R = 0.778$, $p = 0.02$, Fig. 2d, EYA3 upstream TSS: $R = 0.308$, $p = 0.46$, Supplementary Fig. 2e, f). Next, we cloned each EYA3 TSS into luciferase reporters, and using COS-7 cells transfected the reporters along with known E-box regulators[22,23] (see methods for details). This revealed significant activation specific to the downstream (seasonal) TSS (Supplementary Fig. 2g), likely due to the presence of multiple canonical E-box pairs.

**BMAL2 an activator in a circadian-based coincidence timer**. The E-box regulators *CLOCK* and *BMAL1* do not show significant changes in amplitude in the PT under different photoperiods[22,36], therefore we aimed to identify a candidate circadian E-box regulator of the EYA3 downstream TSS. The prediction is that a circadian E-box regulator, would peak in expression only when light falls on the photosensitive phase, as in

the early light phase of LP (approx. ZT4), and would be absent on SP where the photosensitive phase is masked by darkness. The RNA-seq dataset revealed progressive up-regulation by LP of multiple transcripts, and a slower inverse pattern on SP (Fig. 3a, Supplementary Data 3).

Importantly these data identified *BMAL2*, an E-box regulator and paralogue of BMAL1, as progressively up-regulated on LP

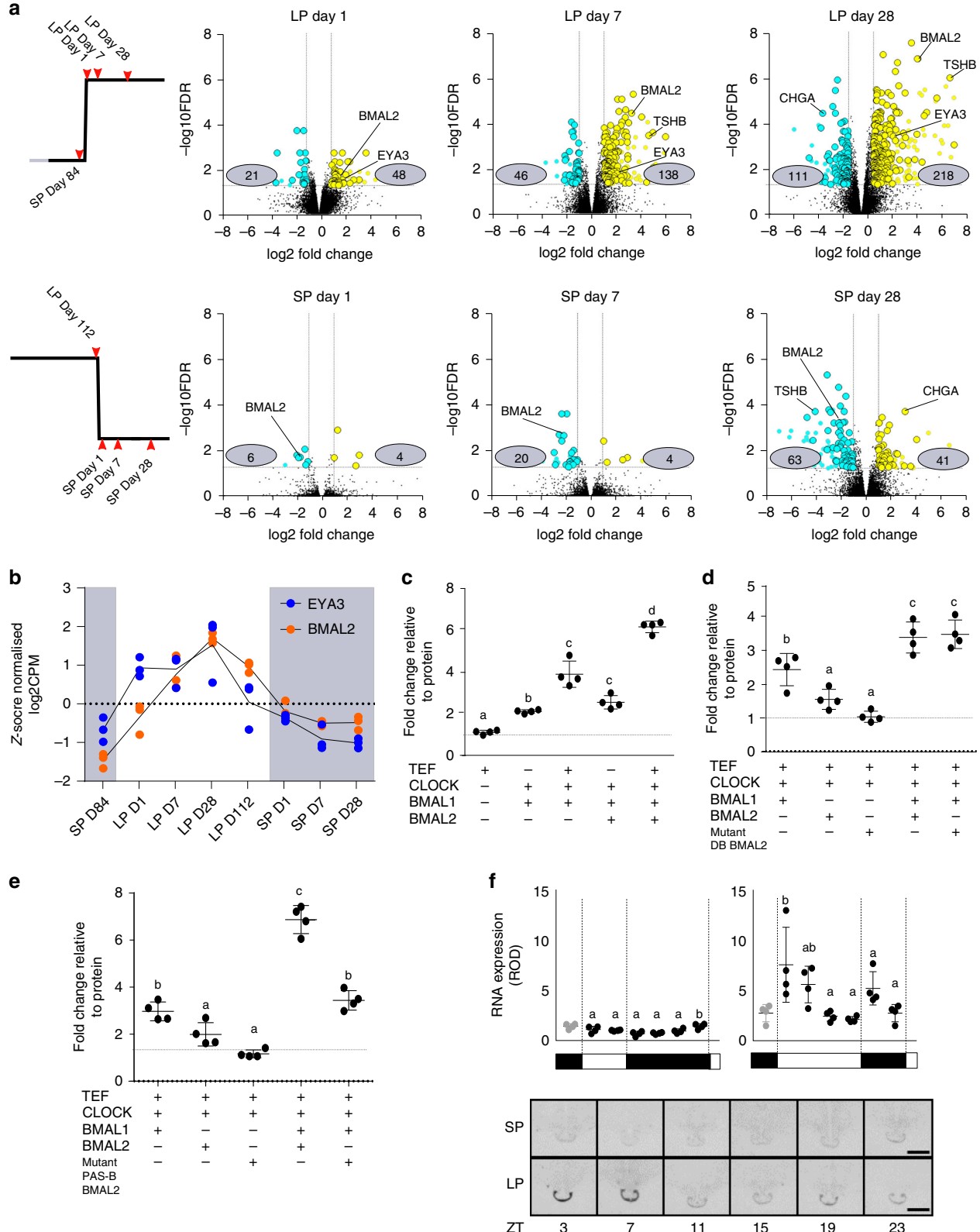

**Fig. 3 Identification of BMAL2 as a seasonally regulated gene. a** Volcano plots showing the number of genes up (yellow) and down-regulated (blue) in pairwise comparisons. The design for the comparisons is shown on the left. These plots demonstrate the consistency of the BMAL2 seasonal signal. Numbers in gray boxes are the differentially expressed genes in the pairwise comparisons (Supplementary Data 3). **b** Z-score normalized $\log_2$CPM RNA-seq plots for EYA3 (blue) and BMAL2 (orange) across the whole experiment. $N = 3$, individual animal values are shown. Statistical significances across the season are found in Supplementary Data 3. **c** Transactivation of the EYA3-downstream-TSS-luc reporter by TEF, CLOCK, BMAL1 and BMAL2. The experiment was repeated 4 times ($n = 4$ per experiment), plot displayed is a representative result. A one-way ANOVA was performed on each individual experiment using Tukey's multiple comparisons test. Different letters indicate significant differences between groups ($p < 0.01$). Error bars SD. **d** Transactivation of the EYA3-downstream-TSS-luc reporter by TEF, CLOCK, BMAL1 and BMAL2. The lack of effect of mutating the DNA binding domain of BMAL2 is shown. The experiment was repeated 4 times ($n = 4$ per experiment), plot displayed is a representative result. A one-way ANOVA was performed on each individual experiment using Tukey's multiple comparisons test. Different letters indicate significant differences between groups ($p < 0.01$). Error bars SD. **e** Transactivation of the EYA3-downstream-TSS-luc reporter by TEF, CLOCK, BMAL1 and BMAL2. The effect of mutating the PAS-B domain of BMAL2 is shown. The experiment was repeated 4 times ($n = 4$ per experiment), plot displayed is a representative result. A one-way ANOVA was performed on each individual experiment using Tukey's multiple comparisons test. Different letters indicate significant differences between groups ($p < 0.01$). Error bars SD. **f** In-situ hybridization and quantification for BMAL2 mRNA from archived material from collected every 4 hours from SP and LP. Representative images are shown ($n = 4$, three independent observations per animal). Error bars represent the SD. Statistical analysis by one-way ANOVA performed, different letters indicate significant differences between groups ($p < 0.01$).

(Fig. 3a, b), closely matching the expression profile for *EYA3* (Fig. 3b). We tested whether BMAL2 was capable of activating the downstream *EYA3* TSS (Supplementary Fig. 2h). In the presence of CLOCK, or alone, BMAL2 had a weak non-significant effect, but in the presence of BMAL1 and CLOCK, we observed significant 4 to 5-fold activation (Supplementary Fig. 2h, Fig. 3c). We then tested whether this depended on direct E-box binding by BMAL2, by mutating the BMAL2 DNA-binding domain (arginine of basic helix-loop-helix domain to alanine R88A[37]), and observed near-identical augmentation of *EYA3* expression (Fig. 3d). The observation that transcription factors can act as co-activators[38] and that bHLH circadian transcription factors require internal PAS domains for functional protein-protein interactions[39], with PAS-B an essential domain for BMAL1 protein-protein interactions[40] led us to mutate the PAS-B domain on ovine BMAL2 (F427R_V439R)[40]. This significantly impaired *EYA3* activation, blocking the co-activation effect with BMAL1 and CLOCK (Fig. 3e). This suggests that BMAL2 operates as a co-activator of *EYA3*, in a CLOCK/BMAL1-dependent manner, requiring PAS-B dependent protein-protein interactions for the mechanism of action, confirmation of this will require suitable antibodies to be raised for future studies.

We assessed the daily profile of *BMAL2* by in-situ hybridization, using archived material[24]. This revealed strong LP-dependent induction of *BMAL2* in the early light phase (ZT3, Fig. 3f), and flat low 24 hour expression on SP. Collectively this reveals that BMAL2 operates as a co-activator driving photoperiodic responses, and fulfils the criteria as a candidate activating arm of the circadian-based coincidence timer.

**Circadian clock repressors shut-down the LP transcriptome.** A prediction from a coincidence timer model is that repressive mechanisms would emerge on short photoperiods in the late night (i.e., photo-inducible phase masked by darkness). To assess this we conducted a serial RNA-seq experiment on PT tissue collected over at 4 hour intervals from both SP and LP-housed animals (Day 28, Figs. 1b and 4a). Transcripts at the two ZT0 collections 24 h apart (ZT0 and ZT24) in each photoperiod had virtually identical RNA-seq profiles, validating dissection technique and down-stream analyses (Supplementary Fig. 3a and Supplementary Data 6). We identified transcripts that significantly changed across the day (diurnal genes) within each photoperiod, and defined the phase, amplitude and period[41] (Supplementary Data 6). More transcripts were rhythmic on SP compared to LP (SP = 880, LP = 643 (Fig. 4b)), with striking asymmetry in the peak phase of expression on both SP and LP (Fig. 4c). Large numbers of genes were expressed on SP in late

night (ZT20, 24) and on LP in the early light phase (ZT4, Fig. 4c), and surprisingly relatively few genes common to both photo-periods (250 transcripts, Fig. 4b, d), but shared genes included the canonical clock genes (Supplementary Fig. 3b).

Amongst diurnal genes shared across photoperiods, dawn-peaking clusters were enriched for the GO term cAMP binding activity, while the dark-onset cluster was marked by the circadian suppressor *CRY1*[42–44] (Fig. 4d, e, Supplementary Fig. 3c). Previous studies have shown *CRY1* is directly induced by onset of melatonin secretion[17,45–47] and that the pattern of induction does not alter between photoperiods. We also observed that regulators of CRY1 protein stability, *FBXL3* and *FBXL21*[22,48,49], are expressed in the PT, but here they marked photoperiod-dependent changes in phasing for *FBXL21* (Fig. 4e).

*BMAL2* emerged as the only transcription factor, which showed both phase and amplitude changes in expression on LP (Fig. 4e, Supplementary Data 6). On LP there was a 4 $\log_2$CPM increase in expression of *BMAL2*, with bi-phasic peaks at ZT4 and ZT20. In contrast, *BMAL1* showed no alteration in amplitude of expression, on either SP or LP (Fig. 4e).

At SP ZT20 we noted a strong functional enrichment for the terms negative regulation of transcription, E-box binding and co-repressor activity (Supplementary Fig. 3c, d). Within this group, we identified 8 genes with known circadian repressor function (*REVERB-α, DEC1, DEC2, CHRONO, FBXL21, KLF10, JUNB, GATA2* and *ERF*, Fig. 4e and Supplementary Fig. 3e). Notably 6 are circadian clock components acting through E-boxes. Cistromic analysis identified over-representation of circadian RORE, D-Box and E-Box sites on the proximal promoter regions in late-night genes on SP (ZT20) and on LP at the dark-light transition (Fig. 4f). Cluster analysis revealed 4 major clusters of E-box motif-enriched genes associated with late night and dawn transition on SP (ZT20) and LP (ZT4) respectively, with *BMAL2* contained within the LP ZT4 cluster (Fig. 4g), and circadian repressor elements within the SP ZT20 peaking cluster (Fig. 4g). This led us to ask whether SP ZT20 repressors more likely to interact, and therefore potentially repress, LP ZT4 up-regulated genes. We used curated experimental protein-protein interaction (PPI) observations from the STRING database, which contain known protein-protein interactions and functional associations[50]. We found that LP ZT4 up-regulated genes are more enriched within the SP ZT20 repressor network ($p$-value = 0.001) than down-regulated genes (Supplementary Fig 4a, b). We did not find significant enrichment when considering genes that are differentially up or down regulated across the whole day $p$-value = 0.25) (Supplementary Fig. 4b). This suggests that SP is defined by an up-regulation of transcriptional repressors responsible for the suppression of the LP activated transcriptome.

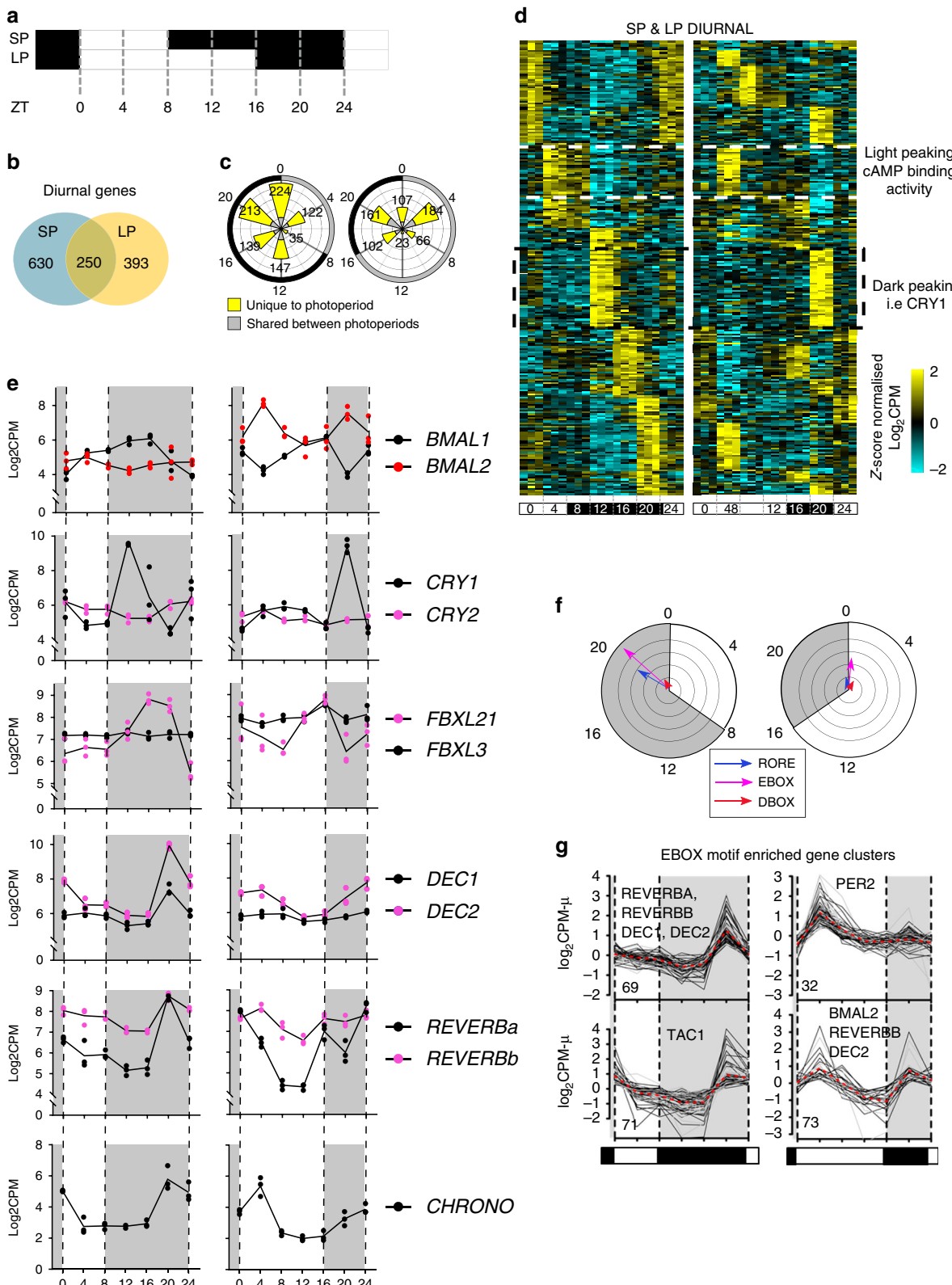

**Melatonin-regulated circadian repression**. Earlier studies have defined the role for melatonin both as an acute inducer of *CRY1*[17,36,45,47], but also determined clear melatonin duration-dependent effects for photoperiodic responses[8,51]. Our data suggest that prolonged SP-like signals may trigger late-night induction of a separate cohort of circadian repressors genes. To test this, we maintained a cohort of animals on LP and then

transferred animals to constant light (LL), a regimen known to suppress the endogenous rise in pineal melatonin secretion[52]. Simultaneously, and at the time of normal lights off (i.e. expected onset of the melatonin signal) we treated animal with an intra-dermal Regulin melatonin implant[47], which mimics the endogenous dark-onset rise of this hormone[53], validated by RIA for melatonin (Supplementary Fig. 5a, *n* = 6). PT tissue was collected

**Fig. 4 Short photoperiods are defined by the expression of transcriptional repressors. a** Experimental design: SP day 28 and LP day 28 the *pars tuberalis* was collected at 4 hour intervals for 24 h. $N = 3$ per time-point. **b** Venn diagram showing the number of diurnal genes (significantly changing throughout the day, FDR < 0.05) in the PT in LP and SP (Supplementary Data 4). **c** Rose plot showing the significant diurnal genes peaking at a particular phase in LP and SP. Gray bars indicate genes shared between photoperiods and yellow indicates they are unique to a photoperiod. Around the outside of the plot the durations of light and dark the animals received is plotted in gray and black respectively, the ZT times are given. **d** Heatmap showing the expression profile of the genes diurnal in both SP and LP (250 genes) for each individual ($n = 3$ per timepoint). Data are ordered by SP peak phase. Yellow are up-regulated and blue down-regulated genes, the data are scaled. Light dark bars and ZT times are given at the bottom of the heatmap. **e** RNA-seq $\log_2$ CPM plots, $n = 3$, individual data shown. Gray shading indicates lights off. Statistical significances are in Supplementary Data 6. **f** Transcription factor binding site analysis by peak phase in SP and LP. Gray and white shading show transition from dark and light. The arrows represent enrichment of either ROREs (blue), E-boxes (pink), or D-boxes (red). The direction of arrows indicate the mean expression peak of genes containing the motif in their promoter and the length of the arrow is the $-\log_{10}$ p-value from a Rayleigh test of uniformity. **g** Cluster analysis of the RNA-seq data identified ZT20 SP peaking and ZT4 LP peaking gene clusters that were enriched for E-boxes. Plots shown are mean normalized $\log_2$CPM of expression profiles to visualize cluster trends. The number of genes in each cluster is indicated in the bottom left. The broken red line represents the most representative medoid gene in the cluster.

for in-situ hybridization analysis at +1.5 h (ZT17.5), +3.5 h (ZT19.5), +6.5 h (ZT22.5) and +9.5 h after hormone treatment, the latter time point being equivalent to 1.5 hours into the pre-dicted light-onset phase on LP (i.e., ZT1.5; Fig. 5a, $n = 4$).

CRY1 expression in the PT was completely suppressed by constant light, but rapidly induced by melatonin treatment, as previously observed[17,45–47] (Fig. 5b). By ZT1.5, expression levels had dropped to basal. In LL conditions DEC1, REVERBα and CHRONO were rapidly elevated in the early subjective dark phase, and in an abnormal pattern when compared to endogenous profiles in LP conditions (Fig. 4e vs Fig. 5c–e). In marked contrast, exposure to melatonin initially repressed all 3 genes, which peaked at +9.5 hours, equivalent to ZT1.5 on LP (Fig. 5c–e). For DEC1 and CHRONO, this pattern closely resembled the phase of the endogenous rise of these genes in animals housed in SP photoperiods (Figs. 4e and 5f). From this, we conclude that multiple repressor arms of the circadian clock are direct melatonin targets and act as a read-out of long duration melatonin signals. This defines a molecular basis for a hormone-regulated circadian-based coincidence timer mechanism, capable of discriminating LP and SP responses within the melatonin target tissue.

**DEC1 suppression of BMAL2-mediated EYA3 activation.** DEC proteins are known E-box suppressors of the circadian clock[54,55], so we selected DEC1 for further analysis. Quantitative in-situ hybridization confirmed the temporal pattern of DEC1 expression (Fig. 5f) in the PT, and using immunohistochemistry of short-photoperiod derived PT tissue, we showed co-localization of DEC1 protein to αGSU expressing thyrotroph cells in the PT (Fig. 5g). DEC1 appears to be specific to thyrotrophs, and was not detected in a common cell-type in the PT, the folliculo-stel-late (FS) cells. We cloned ovine DEC1, and then tested action on BMAL2-mediated induction of EYA3. This showed that DEC1 significantly suppressed the action of both CLOCK /BMAL1 (Supplementary Fig. 5b, c) and CLOCK/BMAL1/ BMAL2-mediated expression of EYA3 (Fig. 5h, Supplementary Fig. 5c). We also tested DEC2, which was without effect (Supplementary Fig. 5d). Hence, DEC1 and BMAL2 exert mutually antagonistic effects on EYA3 expression.

**Discussion**
Our study reveals a circadian coincidence timer within the PT, sculpted by the nocturnal melatonin signal that encodes the mammalian photoperiodic response (Fig. 6). Under LP, BMAL2 exhibits a high-amplitude peak timed approximately 12 hours after the onset of the preceding dark phase, coincident with expression of EYA3 in the early light phase[22]. This 12 hour interval from dark onset to the dawn peak is also remarkably close to the critical photoperiod required to activate a long-day

response in sheep[28,56]. We note that both BMAL2 and EYA3 show a second peak on day 28 of LP, however, these peaks are not coincident (ZT20 BMAL2 and ZT15/16 EYA3[24]). Fur-thermore, the second peak in EYA3 expression is absent in early responses to LP[22] indicating that it is potentially not mechan-isitically important in a coincidence model. We show that BMAL2 acts as a co-activator of EYA3 with CLOCK and BMAL1, likely in a complex through the PAS-B domain interaction[57]. Co-activators are recognized as important rapid-response functional integrators of multiple transcription factors, driving distinct biological programs and environmental responses, including adaption to cold, rapid diet change and disease[38]. Within the PT, BMAL2 and EYA3 may therefore operate as a photoperiodic co-activator cascade. It remains unclear how BMAL2 is regulated, and current models for the regulation of BMAL2 by BMAL1 may not be applicable[58]. Our promoter motif analysis does indicate the presence of E-boxes in BMAL2 but this presents a circular argument and further work on the regulation and evolution of BMAL2 function is required.

While we cannot differentiate between an internal or external coincidence model of photoperiodic time measurement in this study, generally the concept of a photoperiodic coincidence timer predicts that on short photoperiods, repressor activity would dominate in the late night. In line with this we show that long-duration melatonin signals elicit repressor gene transcription in the late night, timed approximately 12 hours after dark onset/rise of melatonin (ZT20 SP). Impressively, these repressors are directly implicated in the negative regulation of LP ZT4 induced genes (Supplementary Fig. 4). Amongst these, we show the cir-cadian repressor DEC1[54] blocks induction, by BMAL2, of EYA3. Therefore, DEC1 and BMAL2 act as a circadian flip-flop switch for photoperiodic time measurement (Fig. 6). It remains unclear whether DEC-mediated repression is via a direct action on E-box sites occupied by BMAL2 or indirect, leading to modification of a co-activator complex. The discovery of transcriptional repressors in the latter half of the night on SP contrasts with previously described acute and photoperiod-independent induction of CRY1 by melatonin onset[17,36,45,47]. However, constant light and there-fore CRY1 repression seems to induce the late-night repressor genes, suggesting that these are direct targets of CRY1-mediated repression in the early to mid nocturnal phase. This observation along with altered phasing by photoperiod of the CRY1 protein stability regulator FBXL21[48,49] indicate that the dynamics of protein degradation of CRY1 could play a role in discriminating melatonin signal duration.

Our study reveals a progressive increase in H3K4me3 marks at transcription start sites of seasonally expressed genes, especially in long photoperiods (Fig. 1d). Furthermore, these long photoperiod induced genes are more likely to have multiple transcription start sites marked with H3K4me3 (Fig. 1e and Supplementary Fig. 1g).

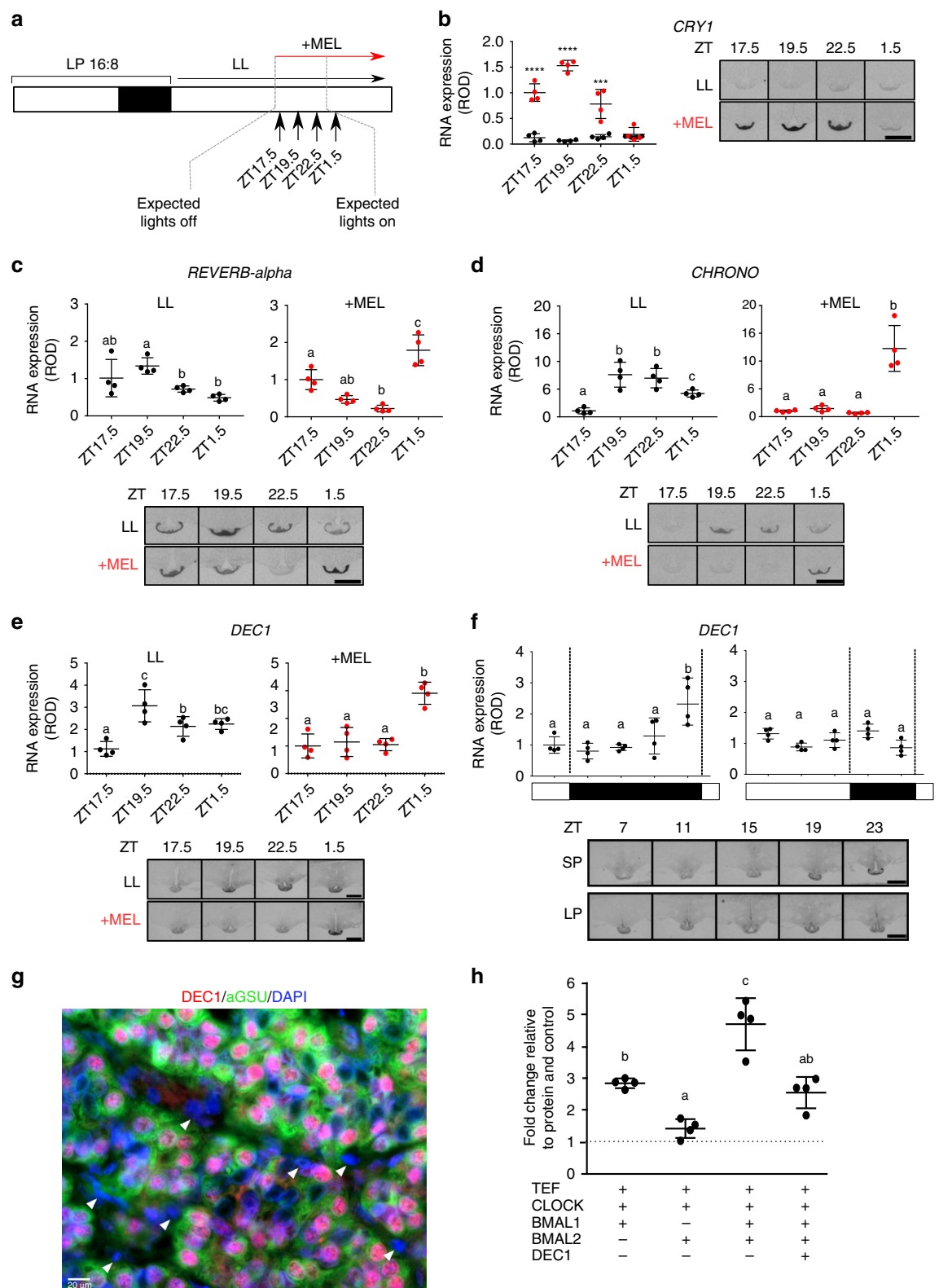

*EYA3* is a particular example of this showing seasonal regulation of only one out of two transcription start sites, presumably seasonally modulating the transcription of EYA3 via BMAL2. On transfer to long photoperiods the *EYA3* and *TSHβ* locus show a progressive increase over a number of weeks in H3K4me3 marks matching the increase in expression (Fig. 2c, d). Our earlier work shows that at the individual cell level the transition between winter and summer physiology is a binary, all-or-nothing phenomenon[29,59]. Integrating these two findings, we suggest that individual thyrotroph cells of the PT exhibit a distribution of critical day length requirements/sensitivity for circadian triggering of the summer physiology leading to a binary switch in cell phenotype, which in a whole tissue assay would appear as a progressive change in epigenetic status.

**Fig. 5 Melatonin duration defines the expression of repressor genes. a** Study design. Animals were sampled at 1.5 hour intervals with either a sham implant or melatonin implant. LL = constant light. **b** In-situ hybridization and quantification for *CRY1* mRNA. Red bars are the value in the presence of melatonin, black bars are in constant light with a sham implant ($n = 4$). Representative images are shown. Statistical significances from a one-way ANOVA and are between the melatonin and sham implant at each time-point. Error bars represent the SD. *p*-value; *<0.01, **<0.001, ***<0.0001, ****<0.00001. **c** In-situ hybridization and quantification for *REVERb-alpha* mRNA. Red bars are the value in the presence of melatonin, black bars are in constant light with a sham implant ($n = 4$). Representative images are shown. Statistical analysis by one-way ANOVA performed, different letters indicate significant differences between groups ($p < 0.01$). Error bars represent the SD. **d** As in c for *CHRONO* mRNA. **e** As in c for *DEC1* mRNA. **f** In-situ hybridization and quantification for *DEC1* mRNA from archived material from collected every 4 hours from SP and LP ($n = 4$). Representative images are shown. Error bars represent the SD. Statistical analysis by one-way ANOVA performed, different letters indicate significant differences between groups ($p < 0.01$). **g** Double immunohistochemistry for DEC1 (red), α-GSU (green) and dapi nuclear stain (blue). White arrow heads indicate FS cells. Scale bar 20 μm. **h** Transactivation of the EYA3-downstream-TSS-luc reporter by TEF, CLOCK, BMAL1 and BMAL2 is repressed by DEC1. The experiment was repeated 4 times ($n = 4$ per experiment), plot displayed is a representative result. A one-way ANOVA was performed on each individual experiment using Tukey's multiple comparisons test. Different letters indicate significant differences between groups ($p < 0.01$). Error bars SD.

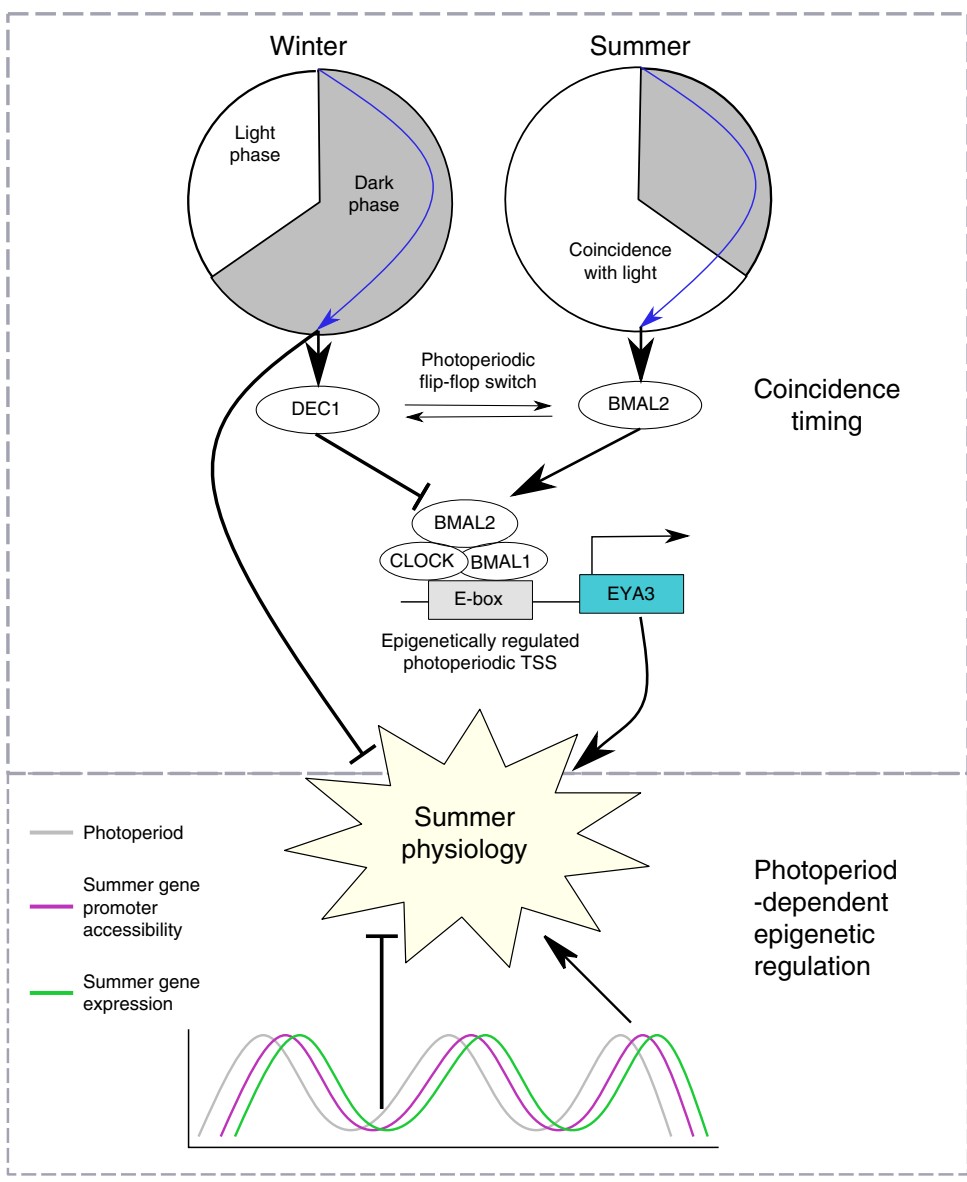

**Fig. 6 Mammalian photoperiodism model: coincidence timing and epigenetic regulation.** The photoperiod sculpts the duration of the melatonin signal, long duration on short photoperiods (SP, winter) and short duration on long photoperiods (LP, summer). In the *pars tuberalis* approximately 12 hours after the onset of darkness there is a photosensitive phase. When this phase coincides with light BMAL2 is expressed. When this phase is coincident with darkness DEC1 is expressed. This forms a photoperiodic flip-flop switch between two stable states. The role of BMAL2, CLOCK and BMAL1 is to co-activate EYA3 and subsequently seasonal physiological changes. DEC1 suppresses the E-box based activation of EYA3. Also occurring in the dark phase of SP a number of repressors are expressed which target appear to target LP induced genes potentially inhibiting the expression of summer physiology. Aside from this acute photoperiodic flip-flop switch there is a progressive photoperiod dependent epigenetic regulation over a number of weeks augmenting the coincidence based timer drive on summer physiology.

We therefore envisage a photoperiodic time measurement model in which a melatonin-regulated circadian-based interval timer interacts with the underlying cellular chromatin state, rhythmically recapitulating the final developmental stages of a thyrotroph endocrine cell in response to long days, leading to an adaptive summer-like physiological state. Our data do indicate morphological changes in another PT cell type, the FS cell, however these cells lack melatonin receptors[60,61] therefore a role in a coincidence timer is unclear. Defining how a flip-flop circadian timer (Fig. 6) interacts with the chromatin state at the level of the single PT cell remains a future challenge.

In summary, the use of photoperiod to synchronise life history cycles in a variable environment is an ancestral feature observed in a large majority of species. Even amongst long day and short day breeders the EYA3-TSH circuitry behaves similarly, demonstrating that although the downstream reproductive responses to photoperiod are altered the mechanism of photoperiodic time measurement is shared[9,13]. The EYA3-TSH circuitry is conserved amongst vertebrates[9,13,16,21], as is the use of a circadian/coincidence based system for photoperiodic time measurement[2,4–7]. Here, we have defined how photoperiodic changes in EYA3/TSH expression stem from a molecular circadian coincidence timer mechanism. We expect this model to be widely applicable across the vertebrate linage.

## Methods

**Animals and experimental design**. All animal experiments were undertaken in accordance with the Home Office Animals (Scientific Procedures) Act (1986), UK, under a Project License held by A.S.I.L. Scottish blackface castrate males were housed in artificial light dark cycles, either 8:16 hour light/dark cycles for short photoperiod (SP) or 16:8 hour light/dark cycles for long photoperiod (LP).

Two separate photoperiod controlled studies were undertaken; 1. The experiment presented in Fig. 1b (seasonal comparison) and Fig. 4a (diurnal comparison at day 28), and, 2. The experiment presented in Fig. 5a (melatonin implant study). Animals were blood sampled throughout the study and terminally sampled at the indicated time-points (Figs. 1b, 4a and 5a). The seasonal experiment was designed to take into account the effects of a photoperiodic switch from SP (SP day 84) to LP and the progressive seasonal changes (LP day 1, 7, 28 and 112), followed by return to SP (SP day 1, 7, 28). Animals were terminally sampled at ZT4 at all time-points. The diurnal comparison was conducted on day 28 animals from this study, they were sampled across the day at 4 hourly intervals for 24 hours. The melatonin implant study was a separate experiment on pre-conditioned LP animals (8 weeks).

All animals were killed by an overdose of barbiturate (Euthatal; Rhone Merieux, Essex, UK) administered intravenously. Hypothalamic blocks with the *pars tuberalis* (PT) and pituitary attached were collected for immunohistochemistry ($n = 3$ per group), electron microscopy ($n = 3$ per group), transcriptomics ($n = 3$ per group), in-situ hybridization ($n = 4$ per group) and epigenomics ($n = 2$ per group). For the omic analyses the PT was dissected to minimize inclusion of transition zone and median eminence. The PT samples were snap frozen on dry ice and stored at $-80$C.

**Hormone assays**. Ovine prolactin (oPRL) was measured as in our previous study[29] for 28 animals during the seasonal experiment (Fig. 1b). In brief, a competitive ELISA using purified oPRL (ovine prolactin NIDDK-oPRL-21; AFP10692C; from Dr. A Parlow, NHPP, Harbor-UCLA Torrance CA, USA) and a highly specific rabbit anti-ovine prolactin (ASM-R50, produced by ASM) were used. Plates were read at 450 nm. The coefficient of variation for the assay on control plasma samples was <10%.

Ovine melatonin was measured by radioimmunoassay as previously described[47] for animals in the melatonin implant study (Fig. 5a, $n = 6$ per timepoint). In brief, using a rabbit anti-melatonin antiserum (PF1288; Paris, France) and 2-Iodomelatonin (NEX236050UC; PerkinElmer, Boston, Massachusetts) as tracer RIA was performed. All samples were assayed in a single assay with an intra-assay coefficient of variation of 5% and a sensitivity of 5 pg/mL.

**Immunohistochemistry**. Tissues ($n = 3$ per group) were immersed in Bouin's fixative for 8 hours, transferred to 70% ethanol, then dehydrated and embedded in paraffin wax. Sagittal sheep brain sections were cut from paraffin embedded tissue at 5 µM, floated onto Superfrost Plus slides (J1800 AMNZ, Thermo scientific), dried at 50 ºC overnight, then dewaxed and rehydrated. Triple immunoflourscence for TSHβ, CHGA and EYA3, and the double immunoflourscence for DEC1/α-GSU was performed as in our previous study[29].

DEC1 primary antibody (CW27, gift from Prof. Adrian L Harris, Weatherall Institute of Molecular Medicine, John Radcliffe Hospital, Oxford)[62] was used at 1:2000. 1:1000 diluted horse radish peroxidase conjugated chicken anti-rabbit IgG antibody was used as a secondary antibody (PI-1000, Vector Laboratories). TSA Plus Cyanine 5 (NEL763001KT, Perkin Elmer) was used to visualize. α-GSU immunofluorescence was used at 1:2000 diluted, ASM-HRSU, R20) and treated with TSA Plus Fluorescein (NEL741001KT, Perkin Elmer). Nuclei were stained by Hoechst 33258 (ab228550, abcam) and cover glasses were mounted by VectaMount AQ (H-5501, Vector Laboratories). Images were collected on a Zeiss Axioimager. D2 upright microscope using a 40x / 0.7 Plan neofluar objective and captured using a Coolsnap HQ2 camera (Photometrics) through Micromanager Software v1.4.23. Images were then processed using Fiji ImageJ (http://imagej.net/Fiji/Downloads).

**In-situ hybridization and quantification of signal**. The OaTSHβ plasmid (XM_004002368.2) was kindly provided by David Hazlerigg. The OaEya3 plasmid (NM_001161733.1) was cloned as previously described[22]. The OaCHGA, OaDEC1, OaREVERB-α, CHRONO and OaBMAL2 were cloned as 1,948-1,970 of XM_004017959.4, 407-767 of NM_001129741.1, 1,012-1,411 of NM_001131029.1, 83-532 of XM_027974329.1, and 1,518-1,906 of XM_027965976.1, respectively.

Frozen coronal ovine hypothalamic blocks ($n = 4$ per group) for in-situ hybridization were cut into 16 µm sections using a cryostat (CM3050s Leica Microsystems, Ltd., Milton Keynes, UK), and thaw mounted onto poly-l-lysine coated slides (VWR International, Lutterworth, UK). Radiolabelled cRNA riboprobes were prepared by plasmid linearization and transcribed using P33 α-UTP (Perkin-Elmer). Fixed sections were hybridized overnight at 60 ℃ with $5 \times 10^5$ cpm of probe per slide. Hybridization signals were visualized on autoradiographic film (Kodak Biomax MR Films, Kodak, USA) after one week exposure at $-80$ ℃. Signal intensity was quantified by densitometry analysis of autoradiographs using the image-Pro Plus 6.0 software (Media Cybernetics, Inc., Marlow, UK).

**Tissue processing and electron microscopy (EM)**. Hypothalamo-pituitary tissue blocks were fixed by immersion in 3% paraformaldehyde/0.05% paraformaldehyde in 0.1 M phosphate buffer (pH 7.2) for 24 hours at room temperature and transferred to a 1:10 dilution of the fixative in 0.1 M phosphate buffer for storage at 4 ℃ before processing ($n = 3$ per group). Using a scalpel blade, areas from the medial PT and median eminence were cut into 0.5 mm³ pieces which were then stained with osmium (1% in 0.1 M phosphate buffer), uranyl acetate (2% w/v in distilled water), dehydrated through increasing concentration of ethanol (70 to 100%), followed by 100% acetone and embedded in Spurr's resin (TAAB laboratory equipment, Aldermarston, UK). Ultrathin sections (50–80 nm) were prepared using a Reichart-Jung Ultracut ultramicrotome and mounted on nickel grids (Agar Scientific Ltd., Stanstead, UK). Sections were then counterstained with lead citrate and uranyl acetate and examined on a JOEL 1010 transmission electron microscope (JOEL USA Inc., Peabody, MA, USA). Sections from 3 animals per group were examined.

For analysis of PT cell morphology, twenty micrographs per animal ($n = 3$ sheep per group) of individual PT cells were taken at a magnification of ×5000. Negatives were scanned into Adobe Photoshop CS2 (Adobe Corp., San Jose, CA, USA) and analyzed using Axiovison version 4.5 (Zeiss, Oberkochen, Germany) image analysis software. The analyst was blind to the sample code. For measurement of the cell and nuclear areas, margins were drawn around the cell or nucleus respectively and the area was calculated. All morphometric values represent the mean ± SEM ($n = 3$ sheep per group). Means were compared by one way analysis of variance (ANOVA) with post hoc analysis by the Bonferroni test. $p < 0.05$ was considered statistically different.

**RNA-seq**. RNA was extracted from the *pars tuberalis* from the seasonal experiment (including the diurnal samples) (Figs. 1b and 4a) using Qiagen's TissueLyser II and RNeasy tissue kit ($n = 3$ per group). The quality of the extracted RNA was assessed using the Agilent 2100 Bioanalyser; all RNA integrity numbers (RINs) were above 8, indicating that good quality RNA had been extracted. Poly-A selection was used.

RNA was prepared with TruSeq Stranded mRNA Sample Preparation Guide, (15031047 Rev. E, Oct 2013) and sequenced on HiSeq 2500 with 125 base pair paired-end reads by Edinburgh Genomics.

The FASTQ files were trimmed with TrimGalore v0.4.0 and mapped (TopHat[63] v2.1.0 and Bowtie[64] v2.3.5) to the 5th release of the sheep genome (Oar_rambouillet_v1.0; assembly GCA_002742125.1). StringTie[65] was used to combine RNASeq and IsoSeq full-length transcripts to generate the genome transcript annotation (accessible in GEO, GSE144677). On average 90% (sdev: 8.46) of paired reads generated were mapped to the genome and 73% (stdev: 0.03) of these were assignable to genes using featureCount[66] (Subread v1.6.3).

All sequence data have been submitted to SRA under the BioProject accession PRJNA391103 and processed data to GEO under GSE144677.

A limma-voom[67] analysis pipeline was used to determine the statistical significance of differential expressed genes. Voom was used to generate normalized precision weighted counts per million (CPM) values which were used in the following regression analyses.

**Seasonal comparison**. The effect of switching from SP to LP was assessed by comparing SP day 84 to LP day 1, 7 and 28, and the effect of switching from LP to SP was assessed by comparing LP day 112 to SP day 1, 7 and 28 (Fig. 1b, Supplementary Data 3). For each gene, we fit a least squared regression model with limma that calculates a single f-test for significance across all model coefficients (mitigating type I errors). Time (days) was treated as a categorical independent variable model for all ZT4 observations in LP and SP (photoperiod × day) in limma and which allowed us to extract from the model the fold change and significance for each pairwise contrasts of interest in limma (Supplementary Data 3; Fig. 3a,b). Significance was determined by an FDR <0.05, >0 $\log_2$CPM abundance and a >1 $\log_2$ absolute fold change.

**Diurnal comparison**. To test for diurnal changes, samples collected at day 28 in LP and SP at a 4 hour time resolution were used ($n = 3$ per group, 7 time-points, Fig. 4a). We used a polynomial regression model approach similar to that of maSigPro[68]. Least squared regression models were then fitted with orthogonal polynomials up to the $5^{th}$ order for time in each photoperiod to identify significantly changing genes. To test for rapid single time-point changes in gene expression a categorical regression model was also fitted to this dataset. Diurnal genes that were significantly changing across time were identified as FDR significance <0.05, $\log_2$CPM >0 and absolute $\log_2$ fold change >1 (Supplementary Data 6; Fig. 4b–d). FDR was calculated throughout using the Benjamini Hochberg correction. Gene expression changes between photoperiods were evaluated by fitting a photoperiod × time (orthogonal polynomials up to $5^{th}$ order) model and extracting the significance and effect size from photoperiod coefficient of the linear model (Supplementary Data 6). In selecting the polynomial we used Akaike information criterion (AIC) to investigate the optimal model selection for expressed genes, balancing model over-fitting and underfitting (using the Oshlack and Gordon selectModel implementation in limma). It is not possible to select a single model that is optimal across all genes, however for genes >0 $\log_2$CPM and with an amplitude >1.5 we found that including orthogonal polynomials up to 5th order was optimal for the most genes in both SP and LP time-series. Again we used the thresholds FDR significance <0.05, $\log_2$CPM >0 and absolute $\log_2$ fold change >1 (Supplementary Data 6). MetaCycle[69] v1.1.0 was used to evaluate gene expression in the 24 h time series for periodicity. JTKCyle[41] and Lomb-Scargle statistics were calculated for an assumed period of 24 hours (Supplementary Data 6). Rayleigh tests for uniformity were performed with the CircStats 0.2-6 package in R. We tested the uniformity of distribution of peak expression times for genes containing each of the core clock motifs (canonical EBOXs (CACGTG), DBOXs (TTA[CT][GA]TAA) and RORE sites (AAN-TAGGTCA)) within the H3K4me3 marked region proximal (within 500 bp) of the TSS.

For all gene cohorts described enrichment analysis of GO terms and pathways was conducted using both consensusPathDB[70] and TopGO[71] 3.1.0. Signicant terms extracted by a FDR <0.05 from a fishers exact test. The weight01 algorithm was used for GO term weighting in TopGO. Where gene annotation did not yet exist for novel and unannotated transcripts OrthoMCL[72] v5 was used to predict orthology from protein sequences and annotation, which were then used to transfer annotation from cow, human, mouse and rat genes. Protein sequences were predicted from novel transcripts using TransDecoder v5.3.0 guided by Uniprot[73] and Pfam[74] best hits to rank coding frames.

We clustered SP and LP time-series profiles using Partitioning Around Medoids (PAM) (Fig. 4g) with the cluster 2.1.0 package in R. Day 28, 24 hour time-series were mean normalized and scaled and PAM clustered with Euclidian distance. The Davies Bouldin index was used to evaluate the optimum number of clusters ($k = 15$ SP and $k = 9$ LP). Motif enrichment of genes clusters was evaluated using fishers two-way exact test against all PT expressing genes as the background. Motifs we identified within H3K4me3 marked regions within 500 bp of a candidate TSS assigned to a gene.

**ChIP-seq**. The method used for ChIP-seq was adapted from an ultra low cell number native ChIP method[75]. In brief, nuclei were isolated from whole PT tissue ($n = 2$ per group) with a dounce homogenizer and sigma nuclear isolation buffer. Mnase digestion was optimized to give the best discrimination between mono, di and tri histones. Immunoprecipitation was performed with protein A and G beads (Invitrogen). Importantly the beads were pre-incubated with the H3K4me3 antibody (active motif) to reduce background noise. Stringent washing with salt buffers was also used to reduce the background noise. The bound chromatin was elute and extracted using a phenol chloroform method. Ampure bead purification was used to clean up the samples. The Qubit and tapestation were used to quantify, a minimum amount of 5 ng was required for the library preparation. The Manchester genomic facility prepared the ChIP-seq libraries and sequenced them according to the standard illumina protocol.

Massive parallel sequencing were performed by Illumina HiSeq4000 with 75 bp paired-end and converted fastqs by bcl2fastq (ver 2.17.1.14). Fastqs were trimmed by trimmomatic (ver 0.36)[76] with the following parameters, ILLUMINACLIP: TruSeq3-PE-2.fa:2:30:10 SLIDINGWINDOW:4:20 MINLEN:35, aligned to the sheep genome (Oar_rambouillet_v1.0) by BWA MEM (ver 0.7.17)[77] with the default parameter and then converted to BAM and sorted by samtools (ver 0.1.19). Peak calling were carried out by MACS2 (ver 2.1.0)[78] with following parameters, -format BAMPE –gsize 2.81e9 -keep-dup 1 -broad -broad-cutoff 0.1 -bdg -SPMR -qvalue 0.05.

Read coverages of peaks were calculated by SICER (ver 1.1)[79] with following parameters; window size = 200, gap length = 200, fdr = 0.01. BED files of replicate samples were merged in order to perform SICER analysis which does not allow replicates. Peaks called by SICER were annotated by HOMER (ver 4.10.3)[80] with default parameters. H3K4me3 peaks identified by SICER were validated by monitoring the distributions on the sheep genome. By HOMER annotation, each peak was described as promoter-TSS (1000 bp from TSSs), exon, intron, TTS, intergenic and the distributions of H3K4me3 peaks were closely resembling to the previous reports[33,34] (Supplementary Fig. 6a). Furthermore, H3K4me3 peaks were well-associated with CpG islands (CGIs) on the sheep genome as described in the previous study (Supplementary Fig. 6b)[35]. We used a standard definition of CpG islands[81]; nucleotides regions with >50% GC content, extending to >200 bp and with an observed vs expected CpG ratio >6.5, and detected them using CgiHunterLight 1.0 on Oar_rambouillet_v1.0 (assembly GCA_002742125.1). H3K4me3 peaks of each sampling day were shuffled by bedtools shuffle (ver 2.27.1)[82] with -noOverlapping as negative controls. For correlation analysis with RNA expression, ChIP read counts of peaks overlapped in ±200 bp from TSSs were used. A workflow diagram can be found in Supplementary Fig. 7.

**ISO-seq**. For gene annotation, five tissue samples were sequenced over two experimental runs using PacBio Iso-Seq. In the first run PT and PD samples were sequenced from an RNA pool of SP and LP Scottish blackface sheep ($N = 1$) and a pineal from a commercial mule sheep from Manchester, UK. This RNA was sent to GATC Biotech (Konstanz, Germany) for cDNA library preparation using their in-house method with mRNA $5'$ cap and poly(A) tail selections and sequencing on a PacBio RSII system. GATC made full length normalized RNA libraries.size selected for <2 kb, 2kb-4kb, >4 kb. sequenced across 75 PacBio RS II SMRT cells [SRX7688275]. In a second run, PT from a pool of sheep in LP, and SP ($n = 3$) were sequenced. RNA was extracted using RNeasy Mini Kit (Qiagen) with on-column Dnase digestion. A full-length cDNA library was constructed for each sample using the TeloPrime Full-Length cDNA Amplification Kit V1 (Lexogen) and amplified using PrimeSTAR GXL DNA Polymerase (Takara Bio) with 22 PCR cycles of 98 °C denaturation for 10 s, 60 °C annealing for 15 s, and 68 °C extension for 10 min. PacBio SMRTbell libraries were prepared using SMRTbell Template Prep Kit 1.0 and each library was sequenced on two SMRT Cells v2 LR using 20-h movies on a Sequel platform at the IMB Sequencing Facility (University of Queensland, [SRX7688271]). All Iso-Seq data was first processed using software IsoSeq v3.1 to obtain full-length non-concatemer reads with at least 3 full sequencing passes, which were then mapped to the sheep reference genome GCA_002742125.1 using GMAP version 2018-05-30. TAMA Collapse from the TAMA tool kit[83] was used to generate unique gene and transcript models, which were further merged with RNAseq-based annotation data using TAMA Merge to incorporate any transcript models that were identified by RNAseq but not Iso-Seq. Functional annotation of transcripts was carried out using Trinotate (v3.1.1).

Where multiple transcripts were present for an expressed gene, with more than one TSS candidate, the active proximal promoter regions were inferred by selecting contiguous H3K4me3 marked regions within 100 bp of a TSS.

**CAGE-seq**. We applied cap analysis gene expression (CAGE) to identify the location and relative expression of TSS regions of the PT across both LP and SP. When combined with IsoSeq and RNASeq derived transcript annotation this provided a comprehensive identification of TSS in the genome which allowed us to more accurately apply DNA binding motif analysis to promoter regions. Libraries were prepared according to Hazuki et al.[32] and sequenced on an Illumina HiSeq 2500 using V4 chemistry on a 50 cycle Single end sequencing run. We sequenced archived RNA samples from the PT in both SP and LP (ZT4, week 12)[29]. We also sequenced RNA from PD (both SP and LP), and Pineal for comparison as out-groups. Reads were trimmed using fastx toolkit 0.0.14 and cutadapt 1.4. Reads were mapped using BWA 0.7.17 to the $5^{th}$ release of the sheep genome (Oar_rambouillet_v1.0; assembly GCA_002742125.1). CAGEr 1.26.0 was used for processing and cluster analysis of TSS (Supplementary Data 4). We filtered reads for a mapping quality >30 and sequencing quality >20. Tag counts were normalized using the power law method with an alpha of 1.12 and T of $10^6$ (determined by plotting the reverse cumulatives of PT samples). We clustering TSSs with >1 tags per million (TPM) together using the distclu methods allowing a max distance between TSS of 20 nucleotides.

**Transcription factor binding site analysis**. Transcription Factor binding motifs were identified using the FIMO tool from MEME v4.11.4 with a p-value threshold of $<1 \times 10^{-7}$. for Jasper 2018 core vertebrate database. Fishers one-way exact tests for enrichment were used to identify the significance of motif enrichment within active promoter regions of a gene cohort compared to a background of all PT expressed genes (>0 CPM). Fishers two-way exact tests were applied to evaluate enrichment and depletion of motifs within active proximal promoter regions of genes with the use of SP and LP H3K4me3 marked regions within 500 bp of candidate TSSs.

**Protein-protein interaction networks**. Using experimentally evidenced protein-protein interaction (PPI) annotation from the STRING[50] v10 database for cow,

sheep, rat, mouse and human we integrated protein interactions in Cytoscape[84] with significantly changing genes at ZT 4 SP vs ZT 20 LP from the 24 hour categorical contrasts, which is 12 hours from dark/melatonin onset (Supplementary Data 6 and Supplementary Fig. 4). A threshold of >0.4 for confidence score in experimental evidence and a combined score of >0.7 was applied, and orphaned proteins removed from the network. The significance of enrichment of PPI repressor connected genes within up-regulated vs down-regulated genes was evaluated using fishers two-way exact tests.

**Cloning and constructs**. Expression plasmids: PCR fragments of the expected sizes were extracted using a gel extraction kit (Qiagen) and cloned in pGEM-T easy vector (Promega); Four to six positive clones were sequenced (MWG, United Kingdom). To generate expression constructs, a second round of PCR was performed using primers flanked by adequate restriction sites and the pGEM-T clone as template. PCR fragments were extracted as described above, digested by the adequate restriction enzymes, purified with a PCR purification kit (Qiagen) and cloned in the expression vector backbone (pCS2-HIS). In order to generate the mutant expression plasmids for BMAL2 we used the QuikChange Lightning Multi Site-Directed Mutagenesis Kit (210515, Agilent). The bHLH mutant was generated by converting an arginine to alanine (OaBMAL2_R88A) based on a mouse mutagenesis study on BMAL1[37]. The PAS-B mutant was made by converting a phenylalanine to arginine, and a valine to arginine (OaBMAL2_F427R_V439R). Based on a on a mouse mutagenesis study on BMAL1[40].

Sanger sequencing of clones are available in Genbank for BMAL2 cds constructs (Genbank: [MT001920]), DEC1 cds constructs (Genbank: [MT019539]), DEC2 cds constructs (Genbank: [MT019540]), PAS-B-mutated BMAL2 cds constructs (Genbank: [MT019541]), bHLH-mutated BMAL2 cds constructs (Genbank: [MT019542]),

Promoter reporter constructs: a strategy identical to that described above was applied and fragments were cloned into the pGL4 basic backbone (Promega) digested with the appropriate restriction enzymes. Sequencing was performed to check accuracy of all re-amplified cloned fragments. EYA3 generic and seasonal promoter construct sequences are available on genebank ([MT001921] and [MT001924] respectively).

**Cell culture, transfection and luciferase reporter assays**. The procedure was as previously reported[22]. In brief, COS-7 cells were grown in Dulbecco's modified eagle's medium supplemented with 10% fetal bovin serum, 1% penicillin/strepto-mycin at 5% $CO_2$ and 37 °C. Cells were plated in 24-well plates at a density of $4 \times 10^4$ cells per ml and incubated for 24 hours prior to transfection. Transfection was performed using Genejuice (Novagen) and the concentration was optmised to transfect the greatest number of cells without compromising cell survival, this was assessed using a luciferase positive control pGL3 containing SV40 (Promega) and trypan blue staining. We recorded a 90% cell survival and a high transfection efficiency. The EYA3 promoter constructs were used at 50 ng per well, as in a previous study[22]. The expression plasmids were used at different doses based on a previous study and optimization of the assay: TEF = 12.5 ng, DEC1 = 25 ng, CLOCK, BMAL1, BMAL2 and mutant BMAL2 were all used at 50 ng, unless otherwise stated. The total transfected DNA amount was set to an equal amount between all conditions by addition of the corresponding empty vector. The luci-ferase assays were performed 48 h after transfection using the luciferase assay kit (Promega) and the Glomax luminometer (Promega). Thetotal protein per well, assessed by Bradford assay was used to normalize the values to total protein content (a proxy for cell number). All data (in Relative Luminescence Units, RLU) represent fold induction once normalized to total protein content and relative to an inert control transfection. Each experiment contained 4 replicate wells and was repeated 4 times giving similar results. An one-way ANOVA using Tukey's mul-tiple comparisons test was performed for each separate experiment conducted in the Graphpad prism 7.05. Representitive plots ($n = 4$) are shown.

## Data availability

All sequence data supporting the findings of this study have been deposited in the Sequence Read Archive (SRA) under the BioProject accession [PRJNA391103] and processed data to GEO under [GSE144677]. Chip-seq raw fastq is available in SRA [SRP110487] and processed bigWig data for in GEO [GSE144515]. Additionally, we have made the bigWig data for rambouillet 1.0 genome [GCF_002742125.1] available as a track hub for use with NCBI browser at https://data.cyverse.org/dav-anon/iplant/home/mhindle/hub.txt that can be loaded in NCBI Genome Data Viewer. CAGE raw data is available in SRA [SRP110487] and as BigWig annotations in the public trackhub. Source data are provided with this paper.

## Code availability

Code used is available from the authors on request.

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

## Acknowledgements

The authors thank Joan Docherty and the staff at the Marshall Building, Roslin, Edinburgh for expert care of their research animals. Past members of the Manchester/Edinburgh team have made contributions to discussions and offered advice, including Sandrine Dupre, and Alex West. We thank David Hazlerigg, Carrie Partch and Jean Michel Fustin for their critical review of earlier drafts of this work. We also thank Dr Gerald Lincoln for his help and assistance over many years. The work was supported by grants from the Biotechnology and Biological Sciences Research Council UK (BB/N015584/1, BB/N015347/1, BBP013759/1) and a Human Sciences Frontier Programme Grant RGP0030/2015 Evolution of Seasonal Timers awarded to ASIL and DWB. ASIL acknowledges the support of the Wellcome Trust, Grant 107851/Z/15/Z.

## Author contributions

S.H.W.—designed the experiments, collected samples, performed RNA preparation, chromatin immunoprecipitation preparation, luciferase reporter assays, cloning and mutagenesis, analyzed/interpreted data, prepared the manuscript and figures. M.H.—bioinformatic analysis, data analysis and submission, sequencing, prolactin assay, figure preparation and revised the manuscript. Y.M.—bioinformatic analysis, immunohistochemistry, luciferase reporter assays, cloning and mutagenesis, figure preparation and revised the manuscript. Y.C.—ISO-seq and bioinformatic analysis. K.M.—immunohistochemistry and analysis. H.C.—performed EM and analysis, revised manuscript. B.R.C.S.—collected samples and performed the in-situ hybridization, luciferase reporter assays, cloning and mutagenesis. N.B.—collected samples and provided lab support. J.M.—developed the novel prolactin assay and prepared antibodies. A.S.M.—collected samples, designed experiments, developed the prolactin assay, and revised the manuscript. S.M.—collected samples, designed experiments, prolactin assay, and revised manuscript. D.W.B.—designed experiments, bioinformatics analysis, and revised the manuscript. A.S.I.L.—conceived the study, designed experiments, collected samples, analyzed/interpreted data, and prepared the manuscript.

## Competing interests

The authors have no competing interests to declare.
