## [Peer Review File · Nature Communications]

Reviewers' comments first round -

Reviewer #1 (Remarks to the Author):

The paper by Wood et al. is a significant and novel piece of work, which provides a plausible mechanistic basis for photoperiodic time measurement in seasonal mammals. Attempting to investigate the molecular basis of seasonal time measurement in (non-standard laboratory) mammals, such as sheep is no easy task but the authors provide some attractive and interesting data in support of a potential mechanism. The mechanism is particularly interesting as it builds on the Bunning hypothesis of external coincidence timing, originally developed for photoperiodism in plants. Furthermore, it suggests how a similar molecular mechanism may explain photoperiodic time measurement in both plants and mammals, pointing to an evolutionarily conserved process and therefore will have broad interest.

Specific comments for consideration by the authors:

1. The paper, and the mechanism put forward, hinges on the importance of two proteins – Bmal2 acting as a transcriptional co-activator, to drive long photoperiod expression of Eya3 and hence downstream TSH and DEC1 as a transcriptional repressor, induced by short photoperiod (long duration night time melatonin) that can block Eya3 expression and hence TSH. The transactivation studies showing the respective activities of these two proteins are quite convincing. Nevertheless, the identification of the two proteins (Bmal2 and DEC1) came from RNA seq studies of differentially expressed genes between long and short photoperiod (Bmal2, Fig 2A; DEC1, Fig 3). While a sound bioinformatic approach was used to narrow in on these two proteins as key players in the photoperiodic timing mechanism, a crucial piece of evidence that is lacking from the study is the demonstration that the Bmal2 and DEC1 are actually found in transcriptional complexes bound to E-boxes in PT tissue. This is important to establish (see below).
2. In Fig 3E, the data to show that Bmal2 changes with both amplitude and phase relative to photoperiod are shown. However, it appears that Bmal1 and Bmal2 are annotated wrongly on the figure, given the description of the data in the text. Should the line marked as Bmal1 be Bmal2 and vice versa?
3. A key element of the proposed mechanism is the role of Bmal2 in driving Eya3 expression and the authors note that Bmal2 peaks at ZT4 and ZT20 on LP. Yet in previous work (Dupre et 2010 (Current Biol. 20, 829–835), the same lab. observed that Eya3 peaks at ZT3 and ZT15 on LP. Thus, there seems to be some desynchrony between the timing of the initial peak in Eya3 and the first peak in Bmal2 (ZT3 v ZT4), which is also evident in this study (in Fig 2b) and a major desynchrony between the second peak in Eya3 (at ZT15) and Bmal2 (ZT20). These observations do not seem to fully match up with the proposed mechanism as Eya3 appears to peak before Bmal2. Also, what is the function of the second peaks in Bmal2 and Eya3 and how do these relate to the proposed mechanism of photoperiodic time measurement. This issue is an issue the authors need to address and hence why it is important to establish the nature of the transcriptional complex in vivo.
4. Another critical piece of information that is how Bmal2 is regulated. What drives the rise in Bmal2 expression? On p6, line 141, the authors use the prediction that any E-box regulator 'would peak in expression only when light falls on the photosensitive phase, as in the early light phase of LP

(approx. ZT4)', from which they identify Bmal2. However, this is unlikely to be direct activation by light, instead it would have to be a light-dependent signal or alternatively it could be a consequence of the removal of short-duration melatonin. Some further insight into how Bmal2 is regulated would help the understanding of the importance of Bmal2 activation in photoperiodic time measurement. 5. In the supplementary Figure 1F, it is shown that the size of the nucleus changes in PT folliculostellate (FS) cells as well as in PT thyrotrophs, yet from this study we know that DEC1 is not expressed in PT FS cells. This suggests that photoperiod driven epigenetic events are not necessarily unique to PT thyrotrophs and begs the questions (a). what role do the PT FS cells play in the photoperiodic timing mechanism, if epigenetic events are important as the authors suggest? and (b) does DEC1 explain all?

Minor point:

1. Figure numbers in the text and on the figures need to be made consistent. Figure 1A, B,C etc are used in the text, but Figure 1a,b,c etc are used on the figures and in the legends.

Reviewer #2 (Remarks to the Author):

Wood et al. present extensive and interesting data from Scottish black face castrated male sheep, a short day breeder, showing genomic and epigenetic correlates of seasonal, photoperiodic timing. They show that photoperiodic changes in the duration of melatonin, which is well established to faithfully reflect the length of the scotoperiod, induce BMAL2 expression when the duration of melatonin is short, which in turn triggers the transcription of EYA3. EYA3 has already been shown to instigate TSHB activity, which increases DIO2 deiodinase activity. Conversely, they show that long durations of melatonin, indicative of the short days of winter induces the repressor DEC1 (as well as DEC2 and REVERBa and b), which suppresses BMAL2 activity and shutting off the EYA3/TSHb process.

The authors indicate that their data support the notion that photoperiodism in the sheep is synchronized by a "circadian coincidence" hypothesis in which the photoperiod entrains a "circadian rhythm of photosensitivity", and the expression of summer or winter biology depends on whether or not light coincides with the phase of high sensitivity". This is specifically the "external coincidence model" of Erwin Bunning (1936), who used this model to explain plant photoperiodism. The authors rightfully cite Bunning's work.

However, in course of their writing, the authors lose the "external" part of the coincidence and merely talk about "coincidence". This may seem to be trivial, but there is another coincidence model. The "internal coincidence model" proposed by Nanda and Hamner (1958) suggested that the same phenomenon could be explained by two circadian oscillators that are differentially entrained to dawn and dusk. Then, as photoperiod changes, the phase relationship between the two oscillators change, inducing or suppressing a seasonal event.

The authors present an interesting set of data in Figure 3 that shows that the phase and amplitude of expression of several "clock genes" or clock-controlled genes as well as occupancy of selected promoter-motifs differentially entrain to the 2 photoperiods. In this reviewer's mind, this is consistent with an internal coincidence model, not Bunning's hypothesis. Frankly, until someone

identifies what is actually being signaled at the "photo inducible phase", I don't think there is a clean way to differentiate the models. Of course, I think it is possible that the expression and phase of expression of EYA3 is in fact the photo inducible phase. If this is the authors' conjecture, they should say so.

I would like the authors to clarify this.

I would also like the authors to clarify the difference and similarity of the epigenetic and genomic changes associated with short day breeders such as sheep and deer with long day breeders such as hamsters and quail. In all 4 cases, DIO2 is induced in long days via similar mechanisms (although quail do not employ melatonin as its proximal photic transducer; their hypothalamus is directly photoreceptive). Yet, in sheep and deer, breeding is suppressed, while in hamsters and quail, breeding is activated.

This is a short paper. I don't expect a long treatise, but I do think the paper would be greatly strengthened by mention of these two different interpretations.

Vincent Cassone

Reviewer #3 (Remarks to the Author):

Attached as 'comments for author'

Reviewer #4 (Remarks to the Author):

I am afraid I am not an expert in circadian biology so I can only comment on the epigenetic part of the paper.

The analysis of H3K4me3 is potentially interesting and could illuminate the story but at present is underdeveloped. I have the following suggestions:

1. H3K4me3 mostly marks CpG island promoters of expressible genes, not necessarily expressed genes. No causality between H3K4me3 and transcription (or the reverse) has been properly established so this should be reflected in the language that's being used. H3K4me3 is not a measure of openness of chromatin; you would have to measure this directly by ATAC-seq or similar methods. So one cannot make any comparisons between H3K4me3 measurements and those of nuclear size or chromatin density. In fact ATAC-seq would have been an excellent thing to do and I'm also surprised the authors haven't done any H3K27me3 measurements, especially as they evoke similarities with plant vernalisation.
2. Please establish the relationship between H3K4me3 and CpG islands in your datasets. For this (and any other) comparison segment your gene classes into non-expressed, constitutively expressed, and seasonally changing (in one direction or other).
3. Please examine all seasonally changing genes (with the controls as above) as to their relationship

with H3K4me3 density. Basically as I understand the hypothesis, seasonal genes show an association with ups and downs in K4 methylation, so this needs to be explored thoroughly not just with example genes. It would also be useful to check if K4 methylases and demethylases are seasonally expressed or not.

4. The choice of timepoints for the experiments needs to be explained better so that a nonexpert reader can understand. Fig 1c please explain the choice of timepoints.

5. There doesn't seem to be any SP D28 in Figure 1 g?

6. Please refrain from making parallels with vernalisation in plants as I believe there is no evidence to suggest there are any.

Response to reviewers: NCOMMS-20-01753 - Circadian clock mechanism driving mammalian photoperiodism

AUTHOR RESPONSES TO IN RED

We thank the 4 reviewers for their evident care to detail, and comments. Each reviewer offered helpful and constructive suggestions, and as a result we feel that the paper has been significantly improved by their input. For this we are grateful.

Reviewer #1 (Remarks to the Author):

The paper by Wood et al. is a significant and novel piece of work, which provides a plausible mechanistic basis for photoperiodic time measurement in seasonal mammals. Attempting to investigate the molecular basis of seasonal time measurement in (non-standard laboratory) mammals, such as sheep is no easy task but the authors provide some attractive and interesting data in support of a potential mechanism. The mechanism is particularly interesting as it builds on the Bunning hypothesis of external coincidence timing, originally developed for photoperiodism in plants. Furthermore, it suggests how a similar molecular mechanism may explain photoperiodic time measurement in both plants and mammals, pointing to an evolutionarily conserved process and therefore will have broad interest.

We thank the referee for the generally positive response to the manuscript and detail a response to specific points below:

1. The paper, and the mechanism put forward, hinges on the importance of two proteins – Bmal2 acting as a transcriptional co-activator, to drive long photoperiod expression of Eya3 and hence downstream TSH and DEC1 as a transcriptional repressor, induced by short photoperiod (long duration night time melatonin) that can block Eya3 expression and hence TSH. The transactivation studies showing the respective activities of these two proteins are quite convincing. Nevertheless, the identification of the two proteins (Bmal2 and DEC1) came from RNA seq studies of differentially expressed genes between long and short photoperiod (Bmal2, Fig 2A; DEC1, Fig 3). While a sound bioinformatic approach was used to narrow in on these two proteins as key players in the photoperiodic timing mechanism, a crucial piece of evidence that is lacking from the study is the demonstration that the Bmal2 and DEC1 are actually found in transcriptional complexes bound to E-boxes in PT tissue. This is important to establish (see below).

We are aware that this is potentially an important issue, and whether the effects we observe are direct or indirect (since BMAL2 may act primarily as a co-activator) is currently unresolved. However, we have provided evidence through luciferase reporter assays that mutation of the DNA binding domain of BMAL2 does not negatively impact its activation potential and that mutation of the PAS/B domain has a detrimental effect, making it less likely that BMAL2 directly binds to E-boxes. The major stumbling block to further progress here is lack of decent antibodies to BMAL2. We have raised a range of different antibodies to different domains of BMAL2, and while these showed some promise in immune-histochemical studies, we have so far been unable to develop successful protocols for their use on co-immunoprecipitation studies. Direct E-box binding of DEC1 has been previously

demonstrated (Li et al, J Biol Chem. 2003;278(19):16899-907). We recognise that here we have not shown direct binding to Eya3 but time and new approaches beyond the scope of this paper are required to demonstrate this. We have modified the text of the paper to reflect this issue

Page 12 line 338-341:

“It remains unclear whether DEC-mediated repression is via a direct action on E-box sites occupied by BMAL2 or indirect, leading to modification of a co-activator complex.”

2. In Fig 3E, the data to show that Bmal2 changes with both amplitude and phase relative to photoperiod are shown. However, it appears that Bmal1 and Bmal2 are annotated wrongly on the figure, given the description of the data in the text. Should the line marked as Bmal1 be Bmal2 and vice versa?

We thank the referee for this and apologise for this obvious error and have corrected the figure.

3. A key element of the proposed mechanism is the role of Bmal2 in driving Eya3 expression and the authors note that Bmal2 peaks at ZT4 and ZT20 on LP. Yet in previous work (Dupre et 2010 (Current Biol. 20, 829–835), the same lab. observed that Eya3 peaks at ZT3 and ZT15 on LP. Thus, there seems to be some desynchrony between the timing of the initial peak in Eya3 and the first peak in Bmal2 (ZT3 v ZT4), which is also evident in this study (in Fig 2b) and a major desynchrony between the second peak in Eya3 (at ZT15) and Bmal2 (ZT20). These observations do not seem to fully match up with the proposed mechanism as Eya3 appears to peak before Bmal2. Also, what is the function of the second peaks in Bmal2 and Eya3 and how do these relate to the proposed mechanism of photoperiodic time measurement. This issue is an issue the authors need to address and hence why it is important to establish the nature of the

Panel A – Figure from Dupre et al 2010.

(A) Eya3 shows significant upregulation in the sheep PT in the light phase (ZT4) on day 1 and day 7 of LP exposure. (B) Eya3 shows low levels of expression in the PT under SP conditions with no significant variation over 24 hr.

The inset shows expression at ZT3. (C) On day 28 of LP, sampling over a 24 hr period shows that Eya3 expression is biphasic with peaks at the early and late light phase (ZT3 and ZT15). The inset shows expression at ZT3.

transcriptional complex in vivo.

Our previous studies (Dupre et al 2010, Current Biology) reporting the photoperiodic regulation of EYA3 were based on different 24h sampling intervals (ZT3,7,11 etc). Archived samples from the Dupre study were used in the current study to validate the BMAL2 RNA-seq result (see figure 2F), this clearly demonstrated a similar time course of induction as seen in Eya3 in the Dupre study. Also in the Dupre et al study

*we included ZT4 profiles for Eya3, with a peak at ZT4 (Panel A above). Generally therefore our earlier data are entirely consistent with the observations we report here, and show an "early morning" peak in expression of EYA3 between 3-4h in duration.*

*Our EYA3 results in the current study also closely match that of the Dupre study for the second peak observation (see Panel B).*

Panel B – RNA-seq from current study. EYA3 expression in LP day 28 (white circles) and SP day 28 (black circles) across the day. ZT time given. ZT0 and ZT24 are equivalent time-points (ie start of light-phase).

*In reference to the "dysynchrony" between BMAL2 and EYA3 second peaks, we concur with the referee that these do not match (see Panel C). However, it is important to note that the EYA3 second peak is only observed at day 28, not earlier in the switch to LP (Dardente et al 2010, day 1, 3 and 7, also in Panel C). This later peak in Eya3 expression "emerges" following prolonged LP exposure and therefore in a coincidence/photoinduction model is potentially not as mechanistically important. The PT is a highly dynamic tissue in a state of constant change, and there is no such thing as a specific "long-day" state but in terms of a coincidence model we must focus on the initial events on LP transfer, which leads to the response cascade. Unfortunately, we do not have 24 hour profile data for BMAL2 from other LP time-points to test this (ie Day 7, 10 etc after LP-exposure) but given the data this seems the most parsimonious interpretation.*

Panel C – Top - RNA-seq from current study showing EYA3 (blue) and BMAL2 (orange) expression on **LP day 28** across the day. ZT time given. Bottom – Figure from Dardente 2010 showing LP day 1, 7 and 15 across the day, EYA3 shown in orange. ZT time given,

4. Another critical piece of information that is how Bmal2 is regulated. What drives the rise in Bmal2 expression? On p6, line 141, the authors use the prediction that any E-box regulator ‘would peak in expression only when light falls on the photosensitive phase, as in the early light phase of LP (approx. ZT4)’, from which they identify Bmal2. However, this is unlikely to be direct activation by light, instead it would have to be a light-dependent signal or alternatively it could be a consequence of the removal of short-duration melatonin. Some further insight into how Bmal2 is regulated would help the understanding of the importance of Bmal2 activation in photoperiodic time measurement.

We know that this is an important issue, but suggest based on our current studies that this may reside outwith the range of the current manuscript. Currently we are undertaking evolutionary and structural analyses to better understand the regulation and function of BMAL2. Below we summarise this work to illustrate the complexity of the task ahead. In reference to the manuscript we have updated the discussion to include reference to this important issue:

Page 12 line 329-332: “Our promoter motif analysis does indicate the presence of E-boxes in BMAL2 but this presents a circular argument and further work on the regulation and evolution of BMAL2 function is required”.

Our unpublished studies on BMAL2 by way of background:

We have looked at the structure of this regulatory region across all vertebrate lineages for which sequence data are available. The general mammalian picture is exemplified by Primates and Ruminants showing 2 conserved E-Box sites in the proximal promoter region. Strikingly in the Muridae rodents, this pattern is broken with an additional site inserted.

Panel D – EBOX-EBOX' pairs are present in BMAL2 promoter regions across the vertebrate lineage and have syntenic conservation within primates and ruminants. Rodents by contrast have multiple additional EBOXs and do not exhibit the typical syntenic conservation even at relatively short evolutionary distances (e.g. across ~5 MY between Mus caroli and Mus musculus).

Panel E – BMAL2 sequence conservation that falls below 90% is highlighted in red for mammalian and rodent sequence alignments. SwissModel was used for structural homology modelling of BMAL2 sequence against the BMAL1 structure.

Additionally, we have collaborated with colleagues at the University of California in some aspects of structural biology studies of BMAL2. These are ongoing, but show that BMAL2 has undergone a substantially different evolutionary path to the “principal circadian regulator BMAL1”. There is marked lack of conservation of multiple domains, with the marked exception of the N-terminal “G-region”. Strikingly, the rodents appear as an “out-group” with a marked loss of selection. Impressively, there is greater sequence variation in BMAL2 between rodents and other vertebrate lineages than there is between Coelacanth and Primates. A current (un-tested) hypothesis is that the conserved G-region may significantly contribute to photoperiodism. This protein has clearly taken a markedly different evolutionary path to BMAL1, or indeed other clock genes.

5. In the supplementary Figure 1F, it is shown that the size of the nucleus changes in PT folliculo-stellate (FS) cells as well as in PT thyrotrophs, yet from this study we know that DEC1 is not expressed in PT FS cells. This suggests that photoperiod driven epigenetic events are not necessarily unique to PT thyrotrophs and begs the questions (a). what role do the PT FS cells play in the photoperiodic timing mechanism, if epigenetic events are important as the authors suggest? and (b) does DEC1 explain all?

FS cells may play an important role in pituitary endocrine regulation, and indeed we reported such an effect in an earlier paper on seasonal prolactin regulation (Dupre et al 2010). Here, we have focused on the EYA3/TSH circuitry, which is exclusively a property of the PT thyrotroph. We do not discount that some of our seasonal changes in RNA and histone modification could be present in the FS cells. Previously we have shown that there are extensive zona adherens, desmosomes, and gap junctions between FS and thyrotope cells in the PT (Wood et al 2015) and previous studies have shown interactions between FS and other pituitary cell types (Baes et al 1987 Endocrinology 120, 685–691; Allaerts et al 1990. Mol. Cell. Endocrinol. 71, 73–81). Thus changes in protein expression in the TSH cells may be transmitted to FS cells via these junctions, but the factors involved will require further investigation, outwith the scope of the present paper. Therefore with specific reference to the reviewer’s comments: (a.) FS do likely have an important role in PTM and our epigenetic and RNA-SEQ results will have included FS cells, meaning we cannot exclude their importance. However, in terms of a cascade of events from photoperiodic change to physiological output the PT thyrotrophs contain the

melatonin receptors, and are therefore the focus on the cell type receiving the photoperiodic signal is logical. (b.) DEC1 is certainly a key component of PTM and appears to be directly responsive to melatonin, fitting with its PT thyrotroph specific expression, and appears to suppress Eya3 activation linking it to the well-characterised TSH circuitry. We do not suggest however that Dec1 regulates all aspects of seasonal physiology; in this study we are focused on the initial events leading to the cascade of photoperiodic events.

Minor point:

1. Figure numbers in the text and on the figures need to be made consistent. Figure 1A, B,C etc are used in the text, but Figure 1a,b,c etc are used on the figures and in the legends.

Done

Reviewer #2 (Remarks to the Author):

Wood et al. present extensive and interesting data from Scottish black face castrated male sheep, a short day breeder, showing genomic and epigenetic correlates of seasonal, photoperiodic timing. They show that photoperiodic changes in the duration of melatonin, which is well established to faithfully reflect the length of the scotoperiod, induce BMAL2 expression when the duration of melatonin is short, which in turn triggers the transcription of EYA3. EYA3 has already been shown to instigate TSHB activity, which increases DIO2 deiodinase activity. Conversely, they show that long durations of melatonin, indicative of the short days of winter induces the repressor DEC1 (as well as DEC2 and REVERBa and b), which suppresses BMAL2 activity and shutting off the EYA3/TSHb process.

The authors indicate that their data support the notion that photoperiodism in the sheep is synchronized by a "circadian coincidence" hypothesis in which the photoperiod entrains a "circadian rhythm of photosensitivity", and the expression of summer or winter biology depends on whether or not light coincides with the phase of high sensitivity". This is specifically the "external coincidence model" of Erwin Bunning (1936), who used this model to explain plant photoperiodism. The authors rightfully cite Bunning's work.

However, in course of their writing, the authors lose the "external" part of the coincidence and merely talk about "coincidence". This may seem to be trivial, but there is another coincidence model. The "internal coincidence model" proposed by Nanda and Hamner (1958) suggested that the same phenomenon could be explained by two circadian oscillators that are differentially entrained to dawn and dusk. Then, as photoperiod changes, the phase relationship between the two oscillators change, inducing or suppressing a seasonal event.

We thank the reviewer for this comment, and we are aware of the distinction in the 2 models. We also admit to shamelessly ducking the issue in the submitted version of the paper, because it is complicated, leading us to use the undefined politically neutral term "co-incidence". However, we have now updated the introduction to highlight the differences:

Page 3 line 53-57: “An “internal coincidence” model has also been proposed where the role of light is to entrain two circadian oscillators and the phase relationship between the two oscillators determines the response to photoperiod 2,3. In either case the role of the circadian clock is central and formal studies support the universality of a “coincidence timer” in animals 2–7, but we lack understanding of the mechanisms involved”.

The authors present an interesting set of data in Figure 3 that shows that the phase and amplitude of expression of several "clock genes" or clock-controlled genes as well as occupancy of selected promoter-motifs differentially entrain to the 2 photoperiods. In this reviewer's mind, this is consistent with an internal coincidence model, not Bunning's hypothesis. Frankly, until someone identifies what is actually being signaled at the "photo inducible phase", I don't think there is a clean way to differentiate the models. Of course, I think it is possible that the expression and phase of expression of EYA3 is in fact the photo inducible phase. If this is the authors' conjecture, they should say so. I would like the authors to clarify this.

Differentiating between internal and external coincidence is a challenge, which is why we have stuck to the neutral “coincidence model”. With the data set from figure 3 we did attempt to assess the phase relationships and protein-protein interactions between the circadian regulated genes in LP and SP to see if we could fit an internal coincidence model. However, these analyses were somewhat inconclusive, and therefore were not included in the manuscript. Our analysis in figure 3 does show a clear shift in peak phase expression (figure 3c & d) but also a very clear peak at ZT4 LP (12 hours after dark) which could be attributed to the presence of a photo-inducible phase (external) or a change in phasing (internal) – but there are a number of unique genes expressed at this time point, coherent with the concept of a release from repression on short photoperiods in the presence of light (external). And indeed we identify SP repressors that target these LP ZT4 induced genes (supplementary figure 4). We have also addressed the complexities internal vs external coincidence timing in the discussion but have not been expansive because this is a nuanced and conceptual view that may not be useful for a wider audience:

Page 12 line 333-334:

“While we cannot differentiate between an internal or external coincidence model of photoperiodic time measurement in this study, generally the...”

I would also like the authors to clarify the difference and similarity of the epigenetic and genomic changes associated with short day breeders such as sheep and deer with long day breeders such as hamsters and quail. In all 4 cases, DIO2 is induced in long days via similar mechanisms (although quail do not employ melatonin as its proximal photic transducer; their hypothalamus is directly photoreceptive). Yet, in sheep and deer, breeding is suppressed, while in hamsters and quail, breeding is activated.

This is a short paper. I don't expect a long treatise, but I do think the paper would be greatly strengthened by mention of these two different interpretations.

This is a well-known issue. The timing system we describe (TSH activation of Dio2) drives a hypothalamic repertoire leading to alteration in reproductive function under environmental control. The reproduce switch occurs upstream of the events in the PT (eya3/tsh) and tanycytes in the 3rd ventricle (DIO2). Ours have suggested and partially demonstrated that the reproductive response in different species is tuned by the effects of “sign-reversal” in down-stream RFamide circuits (review by Angelopoulou E, Quignon C, Kriegsfeld LJ, Simonneaux V. Functional Implications of RFRP-3 in the Central Control of Daily and Seasonal Rhythms in Reproduction. Front Endocrinol (Lausanne). 2019 Apr 10;10:183.). Therefore, so-called long-day and short-day breeding animals show identical TSH/Dio2 responses to LP and we would expect similar epigenetic signatures at the level of the PT. For the reproductive axis, the interpretation of this signal lies down-stream in the hypothalamus. The PT is the primary read-out of photoperiod and the melatonin signal.

While these are interesting points and we do not think a discussion of long day and short day breeders will add to our PT-focused manuscript, and believe other researchers cover this more comprehensively.

Reviewer #3 (Remarks to the Author):

Importance: While some key molecular factors in the regulation of photoperiodicity have been uncovered in the last ~10 years, the mechanism is not well understood. Elucidating this mechanism is important since photoperiodicity drives much of reproductive behaviour in economically important mammalian species. Moreover, uncovering the basic mechanism may well yield insights into how environmental cues interact with molecular signalling pathways that modify a range of physiological processes including reproduction and feeding behaviour. The coincidence model predicts that photoperiod entrains a circadian rhythm of photosensitivity, and that the expression of summer or winter phenotype depends on whether the external light period matches this photosensitive period. Previous work by this group and others have identified the pars tuberalis (PT) of the pituitary gland as a key target organ of pineal melatonin production. The onset of darkness should trigger a circadian oscillator that reaches maximum amplitude during daylight period in summer and night period in winter. Simultaneously, the pineal gland produces melatonin during the hours of darkness. The integration of these steps should output a signal that reaches maximum amplitude in the photosensitive phase in summer, but not in winter. Previous work has demonstrated that the transcription factor EYA3 obeys this behaviour, and is necessary to activate downstream targets of the seasonal response such as bTSH. The outstanding question is: How do circadian oscillator pathways interact with melatonin signalling to modulate EYA3 expression. In order to answer this question the investigators performed a series of temporal transcriptional profiling experiments in order to identify genes encoding transcriptional regulators of EYA3 in the target organ, the ovine PT. Overall, this manuscript represents a considerable step forward in the description of the process of photoperiod regulation. The authors have generated high quality datasets from a well phenotyped model system, and these data will be a considerable resource for the field for years to come. In addition, the authors have identified some new molecular players. However, the paper suffers from a lack of clarity about many of the experimental protocols used, and would be significantly improved by extending the methods

sections and showing more supporting evidence of their experimental protocols. Finally, some conclusions are not supported by the experimental data. My detailed critique is below.

We thank the reviewer for their positive comments and hope we have addressed the issues of clarity in the experimental protocols and that our additional analysis and textual clarifications ensure our conclusions are supported by the data. See detailed response below.

Part 1 (mostly Fig2 and associated datasets).

The investigators clearly justify the timepoints chosen and demonstrate that seasonal endocrine pathways and EYA3 transcription are appropriately dynamically differentially regulated. Next, the transcriptional datasets at these timepoints are interrogated to discover differentially expressed genes that have high expression in the LP but are not induced in the SP. Several genes are identified, but the investigators further refine their investigation to a single target, BMAL2. This is a known Ebox gene that had not been previously implicated in seasonality. In in-vitro assays the team shows that overexpression of BMAL2 can activate the EYA3 promoter in concert with BMAL1, CLOCK and TEF, via the PAS-B domain. Finally, using archival histological material, the authors demonstrate PT – specific induction of BMAL2 gene expression by light in the LP but not the SP.

- The in-vivo methodology and transcriptomics experiment is not well described – crucially, how many replicates were performed per time-point? This should be clearly stated throughout.

We apologise for the lack of clarity and have made the following changes to clarify:

In the results section:

Page 5 line 102: “ (Fig. 1b, Supplementary Fig. 1a-d, n=4)”

Page 5 line 111-113: “Comparing all seasonal time-points (Fig. 1b) we performed ChIP-seq (histone marker H3K4me3, Supplementary Table 1 & 2, n=2) and RNA-seq (Supplementary Table 3, n=3) to screen for seasonal transcriptional activation”.

Page 10 line 277-285: “Simultaneously, and at the time of normal lights off (i.e. expected onset of the melatonin signal) we treated animals with an intradermal “Regulin” melatonin implant 44, which mimics the endogenous dark-onset rise of this hormone 50, validated by RIA for melatonin (Supplementary Fig. 5a, n=6). PT tissue was collected for in-situ hybridization analysis at +1.5 hours (ZT17.5), +3.5 hours (ZT19.5), +6.5 hours (ZT22.5) and +9.5h after hormone treatment, the latter time point being equivalent to 1.5 hours into the predicted light-onset phase on LP (i.e. ZT1.5; Fig. 4a, n=4).”.

In the methods section:

Page 15, line 393 to 405:

“Two separate photoperiod controlled studies were undertaken; 1). The experiment presented in Fig. 1b (seasonal comparison) & Fig. 3a (diurnal comparison at day 28), and, 2). The experiment presented in Fig. 4a (melatonin implant study). Animals were blood sampled throughout the study and terminally sampled at the indicated time-points (Figure 1b, 3a, 4a). The seasonal experiment was designed to take into account the effects of a photoperiodic switch from SP (SP day 84) to LP and the

progressive seasonal changes (LP day 1, 7, 28 and 112), followed by return to SP (SP day 1, 7, 28). Animals were terminally sampled at ZT4 at all time-points. The diurnal comparison was conducted on day 28 animals from this study, they were sampled across the day at 4 hourly intervals for 24 hours. The melatonin implant study was a separate experiment on pre-conditioned LP-housed animals (8 weeks)".
Page 15 line 407-411: "Hypothalamic blocks with the pars tuberalis (PT) and pituitary attached were collected for immunohistochemistry (n=3 per group), electron microscopy (n=3 per group), transcriptomics (n=3 per group), in-situ-hybridization (n=4 per group) and epigenomics (n=2 per group)".

Page 15 line 415-416: "Ovine prolactin (oPRL) was measured as in our previous study 29 for 30 animals during the seasonal experiment (Fig. 1b)".

Page 16 line 422-424: "Ovine melatonin was measured by radioimmunoassay as previously described 44 for animals in the melatonin implant study (Fig. 4a, n=6 per timepoint)".

Page 16 line 430: "Tissues (n=3 per group) were..."

Page 17 line 457: "Frozen coronal ovine hypothalamic blocks (n=4 per group)..."

Page 17 line 473: "...before processing (n=3 per group)..."

Page 18 line 496-498: "RNA was extracted from the pars tuberalis from the seasonal experiment (including the diurnal samples)(Fig1b & Fig. 3a) using Qiagen's TissueLyser II and RNeasy tissue kit (n=3 per group)".

Page 19-20 line 517-562: "Voom was used to generate normalized precision weighted counts per million (CPM) values which were used in the following regression analyses.

Seasonal comparison

The effect of switching from SP to LP was assessed by comparing SP day 84 to LP day 1, 7 and 28, and the effect of switching from LP to SP was assessed by comparing LP day 112 to SP day 1, 7 and 28 (Fig. 1b, Supplementary Table 3). For each gene, we fit a least squared regression model with limma that calculates a single f-test for significance across all model coefficients (mitigating type I errors). Time (days) was treated as a categorical independent variable model for all ZT4 observations in LP and SP (photoperiod x day) in limma and which allowed us to extract from the model the fold change and significance for each pairwise contrasts of interest in limma (Supplemental Table 3; Figure 2a,b). Significance was determined by an FDR < 0.05, >0 log₂CPM and a >1 log₂ absolute fold change.

Diurnal comparison

To test for diurnal changes, samples collected at day 28 in LP and SP at a 4hr time resolution were used (n=3 per group, 7 time-points, Fig. 3a). We used a polynomial regression model approach similar to that of maSigPro63. Least squared regression models were then fitted with orthogonal polynomials up to the 5th order for time in each photoperiod to identify significantly changing genes. To test for rapid single time-point changes in gene expression a categorical regression model was also fitted to this dataset. Diurnal genes that were significantly changing across time were identified as FDR significance <0.05, log₂CPM > 0 and absolute log₂ fold change > 1 (Supplementary Table 6; Figure 3b,c,d). FDR was calculated throughout using the Benjamini & Hochberg method. Gene expression changes between photoperiods were evaluated by fitting a photoperiod x time (orthogonal polynomials up to 5th order) model and extracting the significance and effect size from photoperiod coefficient of the linear model (Supplementary Table 6). In selecting the polynomial we used Akaike information criterion (AIC) to investigate the optimal model selection for expressed genes, balancing model overfitting and underfitting (using the Oshlack

and Gordon selectModel implementation in limma). It is not possible to select a single model that is optimal across all genes, however for genes > 0 log2CPM and with an amplitude > 1.5 we found that including orthogonal polynomials up to 5th order was optimal for the most genes in both SP and LP time-series. Again we used the thresholds FDR significance <0.05, log2CPM > 0 and absolute log2 fold change > 1 (Supplementary Table 6). MetaCycle64 v1.1.0 was used to evaluate gene expression in the 24hr time series for periodicity. JTKCycle 38 and Lomb-Scargle statistics were calculated for an assumed period of 24 hours (Supplementary Table 6). Rayleigh tests for uniformity were performed with the CircStats 0.2-6 package in R. We tested the uniformity of distribution of peak expression times for genes containing each of the core clock motifs (canonical EBOXs (CACGTG), DBOXs (TTA[CT][GA]TAA) and RORE sites (AANTAGGTCA)) within the H3K4me3 marked region proximal (within 500bp) of the TSS.”

Page 21 line 573-580: “We clustered SP and LP time-series profiles using Partitioning Around Medoids (PAM) (Fig. 3g) with the cluster 2.1.0 package in R. Day 28, 24 hour, time-series were mean normalized and scaled and PAM clustered with Euclidian distance. The Davies Bouldin index was used to evaluate the optimum number of clusters (k=15 SP and k=9 LP). Motif enrichment of genes clusters was evaluated using fishers two-way exact test against all PT expressing genes as the background. Motifs we identified within H3K4me3 marked regions within 500bp of a candidate TSS assigned to a gene”.

Page 21 line 583-584: “In brief, nuclei were isolated from whole PT tissue (n=2 per group) with a dounce homogenizer and sigma nuclear isolation buffer”.

We have also updated the ChIP-seq, ISO-seq and CAGE-seq, and transcription factor binding site analysis.

The methods state that the time-course data was fitted to a 5th order polynomial (photoperiod x time), and significant genes generated from this. What is the justification for this method (should be referenced)?

We have added further extensive clarification of our model selection approach within the methods, see above response and the manuscript. To give specifics in response to the reviewer; the polynomial regression approach we used is similar to that within the maSigPro pipeline for time-series analysis (Nueda, et al 2014). We used Akaike information criterion (AIC) to investigate the optimal model selection for expressed genes, balancing model overfitting and underfitting (using the Oshlack and Gordon selectModel implementation in limma). It is not possible to select a single model that is optimal across all genes, however for genes > 0 log2CPM and with an amplitude > 1.5 we found that including orthogonal polynomials up to 5th order was optimal for the most genes in both SP and LP time-series. Panel F shows the number of optimal genes for each model investigated, and is divided into bins for expression ranges to demonstrate how including higher order polynomials becomes more important for the more highly expressed genes.

Panel F - Akaike information criterion for selection of optimal linear/polynomial model for genes

- Other genes (apart from BMAL2) display the predicted pattern of gene expression and could presumably amplify/modulate the EYA3 response. Is it justified for the authors to conclude that BMAL2 is the only important player?

*Our seasonal analysis focuses on ZT4 and identified 48 genes that were up-regulated at the first long day (fig 2a, sup table 3). We then cross referenced to the circadian profile experiment to identify whether these genes showed the appropriate LP ZT4 peak and were defined as changing over the day in LP only (as predicted by a coincidence model)(sup table 6) we found 5 genes, including *eya3* and *bmal2*. *EYA3* and *BMAL2* show a larger fold increase in response to LP than the remaining 3 genes (*ACSL4*, *CREM* and *PTPRN*). The functions of *ACSL4* and *PTPRN* are not compatible with transcriptional activation; *ACSL4* is an enzyme which converts long chain fatty acids. *PTPRN* plays a role in vesicle mediated secretion, which we have noted in earlier work changes with photoperiod (Wood 2015). This leaves *BMAL2* and *CREM* as candidate activators of *EYA3*. *CREM* is a transcription factor which binds to cAMP response elements and has been proposed in the past as signalling “dawn” on long photoperiods (reviewed in Hazlerigg and Loudon Curr Biol. 2008 Sep 9;18(17):R795-R804). However, we failed to detect significant photoperiod-dependent enrichment for cAMP sites in our data set and found that cAMP was related to light responsiveness in both photoperiods (figure 3d). The profile for *CREM* on a seasonal and shows a small low-amplitude oscillation on a daily basis, but substantially less than *BMAL2* (see Panel G). Finally, our unpublished results show that there are no photoperiodic or melatonin-dependent changes in pCREB, suggesting that the cAMP signalling pathway may not be a major factor in photoperiodic time measurement (see Panel G), justifying the focus on *BMAL2*. We felt in an already complicated manuscript it was best not to include this information but if the editor feels it is necessary we are willing to include the data.*

Panel G – RNA expression plots are the seasonal and daily profiles of BMAL2 and CREM from the current study. The IHC images are unpublished results showing on the left pCREB (red) and aGSU (green) and right CREB (red) and pCREB (green). Bar charts in the bottom corner are unpublished quantification of pCREB positive cells after melatonin implant in LP and SP (left) and bracketing dusk and dawn in LP and SP (right) showing no differences.

- The methodology employed for the luciferase assay is not sufficiently described in this paper- instead an earlier work is referred to. The general aspects of the methodology should be described, what was the promoter sequence used? How were the exogenous proteins introduced? There are no data presented to show to what extent the proteins were overexpressed, this should be shown in supplementary information.

We have updated the methods and edited throughout the text:

Page 7 line 168-173: “Next, we cloned each EYA3 TSS into luciferase reporters, and using COS7 cells transfected the reporters along with known E-box regulators 22,23 (see methods for details), this revealed significant activation specific to the downstream (seasonal) TSS (Supplementary Fig. 2g), likely due to the presence of multiple canonical e-box pairs”.

Page 24 – 26 line 698 – 751:

“Cloning and constructs

Expression plasmids: PCR fragments of the expected sizes were extracted using a gel extraction kit (Qiagen) and cloned in pGEM-T easy vector (Promega); Four to six positive clones were sequenced (MWG, United Kingdom). To generate expression constructs, a second round of PCR was performed using primers flanked by adequate restriction sites and the pGEM-T clone as template. PCR fragments were extracted as described above, digested by the adequate restriction enzymes, purified with a PCR purification kit (Qiagen) and cloned in the expression vector backbone (pCS2-HIS). In order to generate the mutant expression plasmids for BMAL2 we used the QuikChange Lightning Multi Site-Directed Mutagenesis Kit (210515, Agilent). The bHLH mutant was generated by converting an arginine to alanine (OaBMAL2_R88A) based on a mouse mutagenesis study on BMAL134. The PAS-B mutant was made by converting a phenylalanine to arginine, and a valine to arginine

(OaBMAL2_F427R_V439R). Based on a mouse mutagenesis study on BMAL1 37.

Sanger sequencing of clones are available in Genbank for BMAL2 cds constructs (Genbank: MT001920), DEC1 cds constructs (Genbank: MT019539), DEC2 cds constructs (Genbank: MT019540), PAS-B-mutated BMAL2 cds constructs (Genbank: MT019541), bHLH-mutated BMAL2 cds constructs (Genbank: MT019542), Promoter reporter constructs: a strategy identical to that described above was applied and fragments were cloned into the pGL4 basic backbone (Promega) digested with the appropriate restriction enzymes. Sequencing was performed to check accuracy of all re-amplified cloned fragments. EYA3 generic and seasonal promoter construct sequences are available on genbank (MT001921 and MT001924 respectively).

Cell culture, transfection and luciferase reporter assays

The procedure was as previously reported 22. In brief, COS-7 cells were grown in Dulbecco's modified eagle's medium supplemented with 10% fetal bovin serum, 1% penicillin/streptomycin at 5% CO₂ and 37°C. Cells were plated in 24-well plates at a density of 4x10⁴ cells per ml and incubated for 24 hours prior to transfection. Transfection was performed using Genejuice (Novagen) and the concentration was optimised to transfect the greatest number of cells without compromising cell survival, this was assessed using a luciferase positive control pGL3 containing SV40 (Promega) and trypan blue staining. We recorded a 90% cell survival and a high transfection efficiency. The EYA3 promoter constructs were used at 50ng per well, as in a previous study (ref). The expression plasmids were used at different doses based on a previous study and optimization of the assay: TEF = 12.5ng, DEC1 = 25ng, CLOCK, BMAL1, BMAL2 and mutant BMAL2 were all used at 50ng, unless otherwise stated. The total transfected DNA amount was set to an equal amount between all conditions by addition of the corresponding empty vector. The luciferase assays were performed 48 hours after transfection using the luciferase assay kit (Promega) and the Glomax luminometer (Promega). The total protein per well, assessed by Bradford assay was used to normalize the values to total protein content (a proxy for cell number). All data (in Relative Luminescence Units, RLU) represent fold induction once normalized to total protein content and relative to an inert control transfection. Each experiment contained 4 replicate wells and was repeated 4 times giving similar results. An one-way ANOVA using Tukey's multiple comparisons test was performed for each separate experiment conducted in Graphpad prism 7.05. Representative plots (n=4) are shown".

- A major experimental conclusion is that BMAL2 is necessary for the induction of EYA3. However, TEF/CLOCK and BMAL1 together appear to have this property without BMAL2. What is the explanation here?

TEF/clock/bmal1 all act on e-boxes therefore can activate eya3. In the artificial system of a luciferase assay we cannot manipulate chromatin state, so the e-boxes are always accessible. In vivo Tef/Clock and bmal1 will always be present in the PT because they have other cellular functions (Lincoln 2002 PNAS 99:21). Therefore, we must view BMAL2's role as a synergistic one with clock and bmal1. The important point here is likely to be the presence of DEC1, an Ebox competitor, on SP, blocking the activation of Eya3 on SP despite the presence of Clock and Bmal1.

- In the methods it is stated for all in-vitro luciferase assays that ‘values are given based on 4 separate experiments with 4 technical replicates per group, therefore n = 16 per group’. This is not so – for the purposes of statistics the number of independent replicates is 4. Stats should be performed with n = 4 throughout.

We have checked and updated the graphs and stats for all figures to show representative plots from one experiment (n=4) and recalculated the statistics on this basis. It has not changed the overall result. Figures that have altered are: Figure 2c, d, e, supplementary figure 2g and h, figure 4h. Note that supplementary figure 5b, c and d have not altered because these are representative plots, we apologise for the oversight and have updated all the figure legends accordingly.

Part 1 overall – the investigators demonstrate that BMAL2 is an important new player in the induction of EYA3 activity, with potential role as part of the key PT molecular driver of photoperiodism. However, experiments should be explained much more clearly, controls (e.g. protein overexpression) shown and stats recalculated with appropriate n.

We have tried to address all the reviewers points above and hope they suffice.

Part 2 (mostly Figure 2 and associated data). Next, the investigators perform experiments to identify the factor that prevents high amplitude expression of BMAL2 in the SP. They reason that such a factor should have high expression during the period that is dark in the winter but light in the summer (ZT8-16). By performing high density temporal transcriptomic profiling the authors generate a high quality, detailed molecular description of the PT in the diurnal cycle in both SP and LP. In ZT20 of the SP functional enrichment analysis identified negative regulators of transcription, notably including E-box regulators. Several enriched genes at ZT4 in the LP contained an E-box in their promoters.

- In Figure 2f the authors show that BMAL2 expression is high in the light period of LP but not SP pituitary. Moreover, they state that BMAL1 expression is not altered by photoperiod. The data in Figure 3e appears to contradict these points. Is the graph miss-labelled?

This was a mistake and the plot was mislabelled.

- The String analysis linking ZT20 repressors and ZT4 targets is poorly described. What is the statistical basis for this analysis? Would any 2 randomly chosen timepoints show a similar level of connectivity, for example? This method should be clearly described and justified. Is there a more rigorous way of showing this interaction?

We have clarified the approach in the methods section and in the results section. Furthermore we have added an additional plot to supplementary figure 4, now 4b. In brief we used the known protein-protein interactions to test whether genes, which are increased in expression at LP ZT4, were more likely to be targets by the products of genes expressed at SP ZT20 (our repressor genes). See the updated text for the results section below:

Page 10 line 258 – 267: “This led us to ask; are SP ZT20 repressors more likely to interact, and therefore potentially repress, LP ZT4 up-regulated genes. We used

curated experimental protein-protein interaction (PPI) observations from the STRING database, which contain known protein-protein interactions and functional associations 47. We found that LP ZT4 up-regulated genes are more enriched within the SP ZT20 repressor network (P -value = 0.001) than down-regulated genes (Supplementary Fig 4a, b). We did not find significant enrichment when considering genes that are differentially up or down regulated across the whole day (P -value = 0.25) (Supplementary Fig. 4b)”.

The methods were clarified, Page 24 line 688, 695-697:

“The significance of enrichment of PPI repressor connected genes within up-regulated vs down-regulated genes was evaluated using fishers two-way exact tests”.

The figure legend was updated for figure 4b:

“b. Number of differentially expressed genes (DEGs), up-regulated (yellow) and down-regulated (blue) for a daily mean (all 24 hour timepoints) between SP vs LP on day 28, compared to LP ZT4 vs SP ZT20 contrast. Connectivity via protein-protein interactions (PPI), defined by STRING to transcriptional repressors is indicated by the checkered shading (also represented in Fig. 4a). The significance of enrichment (fishers two-way exact test) for PPI connectivity within the up-regulated vs down-regulated genes is shown.”

- **Part 2 overall.** The experiments described generate a valuable dataset, but the analysis is poorly described and superficial, without adequate justification.

We have made extensive additions to the explanations of the methodologies used throughout the manuscript, see response to reviewer in point 1 and below:

Page 22-23 line 625 – 651:

“For gene annotation, five tissue samples were sequenced over two experimental runs using PacBio Iso-Seq. In the first run PT and PD samples were sequenced from an RNA pool of SP and LP Scottish blackface sheep ($N=1$) and a pineal from a commercial mule sheep from Manchester, UK. This RNA was sent to GATC Biotech (Konstanz, Germany) for cDNA library preparation using their in-house method with mRNA 5' cap and poly(A) tail selections and sequencing on a PacBio RSII system. GATC made full length normalized RNA libraries.size selected for <2kb, 2kb-4kb, >4kb. sequenced across 75 PacBio RS II SMRT cells (SRX7688275). In a second run, PT from a pool of sheep in LP, and SP ($N=3$) were sequenced. RNA was extracted using RNeasy Mini Kit (Qiagen) with on-column Dnase digestion. A full-length cDNA library was constructed for each sample using the TeloPrime Full-Length cDNA Amplification Kit V1 (Lexogen) and amplified using PrimeSTAR GXL DNA Polymerase (Takara Bio) with 22 PCR cycles of 98 °C denaturation for 10 seconds, 60 °C annealing for 15 seconds, and 68 °C extension for 10 minutes. PacBio SMRTbell libraries were prepared using SMRTbell Template Prep Kit 1.0 and each library was sequenced on two SMRT Cells v2 LR using 20-hour movies on a Sequel platform at the IMB Sequencing Facility (University of Queensland, SRX7688271). All Iso-Seq data was first processed using software IsoSeq v3.1 to obtain full-length non-concatemer reads with at least 3 full sequencing passes, which were then mapped to the sheep reference genome GCA_002742125.1 using GMAP version 2018-05-30. TAMA Collapse from the TAMA tool kit 81 was used to generate unique gene and transcript models, which were further merged with RNAseq-based annotation data using TAMA Merge to incorporate any transcript models that were

identified by RNAseq but not Iso-Seq. Functional annotation of transcripts was carried out using Trinotate (v3.1.1)”.

Page 23-24 line 657 661 & 664-666:

“We applied cap analysis gene expression (CAGE) to identify the location and relative expression of TSS regions of the PT across both LP and SP. When combined with IsoSeq and RNASeq derived transcript annotation this provided a comprehensive identification of TSS in the genome which allowed us to more accurately apply DNA binding motif analysis to promoter regions”” We sequenced archived RNA samples from the PT in both SP and LP (ZT4, week 12) 29. We also sequenced RNA from PD (both SP and LP), and Pineal for comparison as outgroups. Reads were trimmed using fastx toolkit 0.0.14 and cutadapt 1.4. Reads were mapped using BWA 0.7.17 to the 5th release of the sheep genome (Oar_rambouillet_v1.0; assembly GCA_002742125.1). CAGEr 1.26.0 was used for processing and cluster analysis of TSS (Supplementary Table 4). We filtered reads for a mapping quality > 30 and sequencing quality > 20. Tag counts were normalised using the power law method with an alpha of 1.12 and T of 106 (determined by plotting the reverse cumulatives of PT samples). We clustered TSS with >1 TPM together using the distclu methods allowing a max distance between TSS of 20 nucleotides.”.

Part 3 (Figures 3 and 4). In this part the investigators explore the regulation of SP repressors by melatonin, using animals that are kept constantly in light to inhibit melatonin release, and by replacement. The model is validated by demonstrating the expected expression levels of the previously described melatonin-responsive CRY1 gene. Putative repressors identified in Part 2, REVERB-a , CHRONO and DEC1 show a dynamic expression pattern consistent with melatonin regulation. DEC1 was chosen for further analysis since it is a known E-box suppressor of the circadian clock. DEC1 is expressed in the right place for its expected function (thyrotroph cells of the PT). In vitro, overexpression of DEC1 was able to suppress BMAL2-induced EYA3 activation.

- As in part 1, the luciferase experiments are poorly described, expression of overexpressed protein is not shown and the stats is inappropriate.

Please see above response regarding the luciferase assays.

Part 3 overall. This part shows convincingly that REVERB-a, CHRONO and DEC1 are regulated by melatonin and may contribute to the negative regulation of BMAL2 in the SP

Thank you for your positive comment.

Part 4. Epigenetic analyses. In supplement to the transcriptional profiling performed in Part 1, the authors carry out ChIP-seq experiments to examine the binding of the histone modification H3K4me3. This is a mark that is associated with the promoters of actively transcribed genes – indeed, active RNA-PolIII recruits this mark to the promoter in concert with transcription. This mark has also been associated with intragenic regions and enhancer sequences. The authors report coincidence of H3K4me3 binding with gene expression levels, and state that this correlation is ‘stronger on LP indicating a global activation of gene transcription on LP associated with increased chromatin accessibility’.

- The analysis of the ChIP seq dataset is poorly described in the methods and impossible to follow. How is H3K4me3 density called? The authors should clearly present their analysis workflow and extend the methods section to include sufficient information to allow this analysis to be replicated.

The methodologies have been updated to include a better explanation of the analysis completed, we have also included a workflow diagram, see supplementary figure 7 (and Panel H) and page 22 line 605- 623:

“Read coverages of peaks were calculated by SICER (ver 1.1) 74 with following parameters; window size=200, gap length=200, fdr=0.01. BED files of replicate samples were merged in order to perform SICER analysis which does not allow replicates. Peaks called by SICER were annotated by HOMER (ver 4.10.3) 75 with default parameters. H3K4me3 peaks identified by SICER were validated by monitoring the distributions on the sheep genome. By HOMER annotation, each peak was described as promoter-TSS (1000 bp from TSSs), exon, intron, TTS, intergenic and the distributions of H3K4me3 peaks were closely resembling to the previous reports 76,77 (Supplementary Fig. 6a). Furthermore, H3K4me3 peaks were well-associated with CpG islands (CGIs) on the sheep genome as described in the previous study (Supplementary Fig. 6b) 78. We use a standard definition of CpG islands79; nucleotides regions with > 50% GC content, extending to > 200 bp and with an observed vs expected CpG ratio >6.5, and detected them using CgiHunterLight 1.0 on Oar_rambouillet_v1.0 (assembly GCA_002742125.1).H3K4me3 peaks of each sampling day were shuffled by bedtools shuffle (ver 2.27.1) 80 with -noOverlapping as negative controls. For correlation analysis with RNA expression, ChIP read counts of peaks overlapped in \pm 200 bp from TSSs were used”.

In addition we included an additional workflow figure – supplementary figure 7, the ChIP-seq workflow is shown in Panel H.

Panel H – ChIP-seq workflow

- The correlation of gene expression with ‘H3K4me3 density’ is a very unusual way of presenting this kind of data. Generally ChIP seq peaks are called using a sliding window approach to differentiate enriched vs background levels. The peak is subsequently treated as digital – ie, present or absent (rather than a continuous variable). Peak calls are then mapped to reference genome

datasets to identify, e.g, association with transcriptional start sites, introns, repetitive DNA, etc. The authors should show this analysis for their data to confirm i) sequence distribution of their H3K4me3 peaks to different genomic features, ii) overlap of peaks with expressed genes at TSS/elsewhere with a clearly defined window.

We thank the reviewer for their suggestions of additional analyses and have completed these and present the peak call distributions in supplementary figure 6 and Panel I. The distributions of H3K4me3 peaks were closely resembled previous reports and H3K4me3 peaks were well-associated with CpG islands (CGIs) on the sheep genome. For details please see the manuscript at page 22 line 605- 623 (also pasted in above response point 10). We also updated the results Page 5 line 11-121.

We also note that the phrase "H3K4me3 density" was a terminology issue on our part and we have updated the text to either use "peaks" or "marks" throughout.

Panel I – Quality check of H3K4me3 ChIP-seq.
a. Pie charts revealing distributions of H3K4me3 peaks on each genomic feature. Peaks of promoter-TSS were located on ± 1000 bp from TSS.
B. Bar plots revealing percentages of H3K4me3 peaks co-localised with CGIs (left) and CGIs co-localised with H3K4me3 peaks (right) on the sheep genome. Black is observed H3K4me3 peaks in PTs and grey is randomly shuffled peaks with the same fragment sizes as negative controls.

For the correlation analysis we improved our approach and updated figure 1d (see Panel J). Note: We changed the labels of figures to "H3K4me3 peak read counts around TSSs" from "H3K4me3 density". The figure below shows the distributions of Pearson correlation coefficient between gene expression and normalised readcounts of H3K4me3 marks around

Panel J – Histogram revealing frequency distributions of Pearson correlation coefficients between RNA expression (\log_2 CPM) and H3K4me3 peak read counts ± 200 bp from TSSs (\log_2 read counts). Red bars are seasonally expressed genes (\log_2 fold change ≥ 1 or ≤ -1 and adjusted p value < 0.05 of SPday84 vs LPday1, 7, 28 and LPday112 vs SPday1, 7, 28 shown in Fig. 2a, differentially expressed genes (DEGs)=480, duplicates were removed) and black bars are non-seasonally expressed genes (\log_2 fold change ≤ 0.1 and ≥ -0.1 from the same pairwise above, number of genes=218, duplicates were removed).

TSSs. Seasonal genes were identified from differentially expressed genes between SPday84 and LPday1, day7, day28 and between LPday112 and SPday1, day7, day28 (log fold change ≥ 1 or ≤ -1 and adjusted p value < 0.05 , shown in fig 2a, N=480). Non-seasonal genes were non differentially expressed genes from the same pairwises (log2 fold change < 0.1 and > -0.1 , N=281). Peaks of H3K4me3 were identified by using SICER. Overlapping peaks with ± 200 bp from TSSs of seasonal or non-seasonal genes were used for the analysis. Both expression and peak readcounts were log-scaled. The genome-wide analysis also showed the correlations of seasonal expression of RNA and H3K4me3 marks on their promoter regions. However, the majority of non-seasonal genes showed no correlation between seasonal expression and seasonal H3K4me3 peak counts.

We have updated the results section to reflect this:

Page 6 line 122 – 133: “ Next we identified seasonally expressed genes, as defined by RNA-seq analysis of differentially regulated genes (DEGs) in the SP to LP and LP to SP transfers (Supplementary Table 3), and observed a strong correlation between seasonal gene expression and H3K4me3 peaks around the transcription start sites (TSS’s)(Fig. 1d). Importantly, this correlation was absent in non-seasonally regulated genes (Fig. 1d)”.

Page 36-37 line 1037-1045: “Histogram revealing frequency distributions of Pearson correlation coefficients between RNA expression (log2 CPM) and H3K4me3 peak read counts ± 200 bp from TSSs (log2 read counts). Red bars are seasonally expressed genes (log2 fold change ≥ 1 or ≤ -1 and adjusted p value < 0.05 of SPday84 vs LPday1, 7, 28 and LPday112 vs SPday1, 7, 28 shown in Fig. 2a, differentially expressed genes (DEGs)=480, duplicates were removed) and black bars are non-seasonally expressed genes (log2 fold change ≤ 0.1 and ≥ -0.1 from the same pairwises above, number of genes=218, duplicates were removed)”.

- What is the evidence that H3K4me3 density drives chromatin accessibility? In any case, the correlation analysis described in Figure 1c does not show this. It shows rather that genes that are down-regulated in the critical interval are associated with reduced relative H3K4me3 deposition, and genes that are upregulated have increased H3K4me3 deposition. This is entirely expected since PolII activity recruits H3K4me3. The gold standard to show that chromatin accessibility is increased globally in LP (as the authors imply by their inclusion of nuclear density measures (Fig1d and suppl)) is to perform chromatin-conformation capture type experiments, which measure the interactivity of distal parts of the genome.

We have updated the manuscript throughout to clarify that our findings are related to h3k4me3 marks not chromatin accessibility per se. However, the relationship between H3k4me3 marks and chromatin accessibility has been previously reported with h3k4me3 regulating gene expression through chromatin remodelling by the NURF complex making chromatin more accessible for transcription factors (Wysocka Nature. 2006;442(7098):86-90). Obviously it would be interesting to conduct an ATAC-seq experiment to look specifically at accessibility, as our EM data clearly indicates this is a feature of LP and a whole range of histone and chromatin modifiers could be responsible for this. The manuscript has been updated and improved, especially at pages 5 – 6 line 103-150.

- In the discussion the authors state that the progressive increase in H3K4me3 binding at EYA3 over the weeks leading into the LP is at odds with previous data where they show that at the individual level there is a binary switch in gene expression. I disagree – the quantitative increase in H3K4me3 that the authors observe at the TSS of EYA3 could represent a binary H3K4me3 TSS deposition in an increasing number of cells across the time interval. Again, the histone mark is likely to be secondary to the transcriptional activation.

We entirely agree with the reviewer and apologise because we must not have written these statements clearly. We have updated the text as follows, page 13 line 364-370:

“Our earlier work shows that at the individual cell level the transition between winter and summer physiology is a binary, all-or-nothing phenomenon 29,56. Integrating these two findings, we suggest that individual thyrotroph cells of the PT exhibit a distribution of critical day length requirements/sensitivity for circadian triggering of the summer physiology leading to a binary switch in cell phenotype, which in a whole tissue assay would appear as a progressive change in epigenetic status”.

- The authors note the interesting finding that EYA3 appears to have a seasonal TSS. Is this a more generalised phenomenon? The authors should interrogate their dataset to assign H3K4me3 peaks to ‘canonical’ and ‘non-canonical’ TSSs. Should there be multiple instances of this, motif analysis of these sites could potentially uncover seasonality-specific regulatory mechanisms.

We thank the reviewer for pointing this out and have completed the suggested analysis, which is now included in supplementary figure 1g,h (Panel K) and updated the manuscript accordingly:

Page 6 line 134- 144: “We noted that approximately 70% of seasonally DEGs (Supplementary Table 3) had more than one TSS compared to only ~20% of the PT genomic background (Supplementary Fig. 1g, Supplementary Table 4). Next we took genes that were up-regulated in either SP or LP and plotted the proportion of genes with multiple TSS’s, and repeated this for H3K4me3 marked TSS’s, this revealed that H3K4me3 marks are more likely to occur on genes with multiple TSS’s (Supplementary Fig. 1h) and highly expressed seasonal DEGs have a greater prevalence of multiple TSSs than non-seasonal genes expressed at the same level. Furthermore, this phenomenon was more pronounced in LP than SP. This indicates an enrichment of multiple TSS’s in LP up-regulated genes which is associated with the H3K4me3 mark”.

Page 42 line 1217 -1220: “g. Percentage of genes with a given number of transcription start sites in the genomic background (all >0 log₂CPM expressed genes) of the pars tuberalis (grey bars) as compared to all seasonally differentially expressed genes (white bars)”.

Page 42-43 line 1222-1230: “h. Comparison of the prevalence of multiple (>1) TSS across different gene cohorts. The cohorts are LP 28 days up-regulated DEGs (solid red), SP 28 days up-regulated DEGs (solid blue) and all PT expressed genes as the background (solid black). Prevalence of multiple TSS on genes is shown for all thresholds (%) of the uppermost expressed genes (i.e. increasing thresholds for the upper quantile of gene expression). The equivalent

gene expression ($\log_2\text{CPM}$) values for upper quantiles for a lower threshold cutoff upper x-axis. Dashed lines indicate the proportion of gene in the cohorts with multiple H3K4me3 (>1) marked TSS”.

Panel K – g. Percentage of genes with a given number of transcription start sites in the genomic background (all >0 $\log_2\text{CPM}$ expressed genes) of the pars tubercalis (grey bars) as compared to all seasonally differentially expressed genes (white bars). H. Comparison of the prevalence of multiple (>1) TSS across different gene cohorts. The cohorts are LP 28 days up-regulated DEGs (solid red), SP 28 days up-regulated DEGs (solid blue) and all PT expressed genes as the background (solid black). Prevalence of multiple TSS on genes is shown for all thresholds (%) of the uppermost expressed genes (i.e. increasing thresholds for the upper quantile of gene expression). The equivalent gene expression ($\log_2\text{CPM}$) values for upper quantiles for a lower threshold cutoff upper x-axis. Dashed lines indicate the proportion of gene in the cohorts with multiple H3K4me3 (>1) marked TSS.

- **Part4 overall.** This part of the work is the weakest. It does not really increase the informative value of the paper in the current state. Moreover, I feel that the conclusions drawn are not supported by the data. These data could be removed with minimal impact on the overall quality of the work. Should the analysis of seasonal-specific TSS be extended and prove a generalised phenomenon, then this is a significant and interesting new finding.

We believe that this is the first demonstration of genome wide seasonal changes in epigenetic status and therefore is of value to report. We have taken the reviewers useful comments on board and strengthened the data regarding the correlation of seasonal gene expression and h3k4me3 counts at TSSs and demonstrated that the multiple TSS phenomenon is a more generalized feature. We believe these are significant and interesting findings that should remain in the manuscript.

Reviewer #4 (Remarks to the Author):

We thank the reviewer for their helpful comments, much of this reviewers suggested changes and analyses overlapped with reviewer 3, so to avoid repeating information we endeavour to reference to those responses where relevant and note figure numbers and page/line references in the manuscript rather than full copying of text.

I am afraid I am not an expert in circadian biology so I can only comment on the epigenetic part of the paper. The analysis of H3K4me3 is potentially interesting and could illuminate the story but at present is underdeveloped. I have the following suggestions:

1. H3K4me3 mostly marks CpG island promoters of expressible genes, not necessarily expressed genes. No causality between H3K4me3 and transcription (or the reverse) has been properly established so this should be reflected in the language that's being used. H3K4me3 is not a measure of openness of chromatin; you would have to measure this directly by ATAC-seq or similar methods. So one cannot make any comparisons between H3K4me3 measurements and those of nuclear size or chromatin density. In fact ATAC-seq would have been an excellent thing to do and I'm also surprised the authors haven't done any H3K27me3 measurements, especially as they evoke similarities with plant vernalisation.

We have extensively revised the language in the manuscript to be more precise and clear, and a better reflection of the current knowledge on chromatin/h3k4me3, especially at pages 5 – 6 line 103-150. Please see the response to reviewer 3 point 12 for extra information and a response in reference to ATAC.

We did in fact do H3K27me3 measurements but found very little correlation with seasonal transcriptional status and seasonality. This was a disappointment to us considering the plant literature and our own data showing marked seasonal variation in expression of the EZH2 transcript. As these data were largely negative, we chose not to report a large and extensive negative data set. However, if the editor feels this is appropriate we could include these data. We acknowledge that our references to vernalisation "muddies the waters" and have removed direct parallels, only using the argument in the introduction as a means to set up the concept of seasonal epigenetic regulation but any reference to our data showing this has been removed.

2. Please establish the relationship between H3K4me3 and CpG islands in your datasets. For this (and any other) comparison segment your gene classes into non-expressed, constitutively expressed, and seasonally changing (in one direction or other).

We thank the reviewer for their suggestions for additional analyses, this was also suggested by reviewer 3 in point 10 and 11. These have been completed and can be found in supplementary figure 6 and Panel I in this response. The distributions of H3K4me3 peaks were closely resembled previous reports and H3K4me3 peaks were well-associated with CpG islands (CGIs) on the sheep genome. For details please see the manuscript at page 22 line 605- 623. We also updated the results Page 5 line 11-121.

3. Please examine all seasonally changing genes (with the controls as above) as to their relationship with H3K4me3 density. Basically as I understand the

hypothesis, seasonal genes show an association with ups and downs in K4 methylation, so this needs to be explored thoroughly not just with example genes. It would also be useful to check if K4 methylases and demethylases are seasonally expressed or not.

Reviewer 3 (point 11) also encouraged us to improve our correlation analysis and we have update figure 1d, see panel J in this document and our response to reviewer 3. In the manuscript Page 6 line 122 – 133 and Page 36-37 line 1037-1045.

The reviewer's second point relates to the regulators of the h3k4me3 marker. Histone modifications are precisely balanced by methyltransferases ("writers"), demethylatases ("erasers") and effector proteins ("readers"), so we took the known writers, readers and erasers (Hyun 2017) and checked our rnaseq data. We found no seasonal changes in expression. These data are added to Supplementary Table 5 and the manuscript is updated:

Page 6 line 127-133: "Histone modifications are precisely balanced by methyltransferases ("writers"), demethylases ("erasers") and effector proteins ("readers"), therefore we checked the RNA expression of H3K4me3 readers, writers and erasers but found no seasonal changes (Supplementary Table 5). This suggests that changes in protein activity of H3K4me3 modulators may be key in the observed seasonal alterations in H3K4me3 marks.)".

4. The choice of timepoints for the experiments needs to be explained better so that a nonexpert reader can understand. Fig 1c please explain the choice of timepoints.

We have now updated Fig1c (now fig 1d) to include all comparisons to remove a time-point bias, please see response to point 4 above. Furthermore the manuscript has been updated extensively to improve the description of experiments and time-points, see response to reviewer 3 points 1,4,8,9, and 10. We have added supplementary figure 7 to explain the analysis pipeline for ChIP-SEQ and updated the methods: Page 15 line 393-405, page 19 line 519- 562 and 573 – 580. Page 22 line 605 – 676. Page 25 line 707 – 752.

We have also made a number of changes in the results to clarify and explain the analysis better, especially pages 5 – 7 lines 103 - 173.

5. There doesn't seem to be any SP D28 in Figure 1 g?

Figure legend 1g has been updated to clarify the exclusion of SP day 28 and 84. "Note: SP day 28 and 84 are not included because the H3k4me3 peaks are non-detectable".

6. Please refrain from making parallels with vernalisation in plants as I believe there is no evidence to suggest there are any.

We acknowledge that our references to vernalisation "muddy the waters" and have undertaken to remove direct parallels and only use it in the introduction as a means to set up the idea of seasonal epigenetic regulation.

Peer Review File

Reviewers' comments second round -

Reviewer #1 (Remarks to the Author):

The authors have addressed the comments in full and have provided some comprehensive responses. In general terms I am content with the responses provided, but there are couple of minor issues for consideration.

Point 1: It is accepted that as antibodies are not currently available against Bmal2 and therefore the authors are currently unable to resolve the composition of the transcriptional complex. In the light of this the additional statement added to the text provides helpful caution.

Point 2: Error corrected

Point 3: The explanation offered is helpful and accepted. In particular, the authors agree that there is desynchrony in the second peaks in BMAL2 and EYA3 and they offer a logical explanation why this is unlikely to be mechanistically important– namely the second peak in EYA3 is not observed early following switch to LP. However, the authors should also make this clear in the MS, by adding a sentence of explanation to this effect in the discussion.

Point 4: It is accepted that what drives Bmal2 expression is an important issue for future work and is beyond the scope of the current study.

Point 5: The point here is that the authors show data in supplementary Fig 1F, which shows clear photoperiodic changes in the size of the nucleus of the FS cells as well as the PT thyrotrophs. There is no comment or reference to these changes in the FS cells in the results section of the paper, but there is a clear statement that there are no morphological changes in the pars distalis somatotrophs. The authors use this contrast between the PT thyrotrophs and PD somatotrophs to support their argument for increased chromatin accessibility which may be the basis for the photoinductive effects of LP leading to changes in the seasonal transcriptome. While this is reasonable, they should nonetheless refer to the changes in the nuclear size of the FS cells in the results, and not just ignore them. In their rebuttal letter, the authors acknowledge that they cannot exclude the importance of the FS cells, but suggest they are less relevant to the current study on the basis that the PT thyrotrophs contain the melatonin receptors. Therefore, they focus their attention on the thyrotrophs as it receives the photoperiodic signal. While I agree that the PT thyrotrophs are the main output cell, there is no evidence (I know of) that the FS cells do not also express melatonin receptors. Indeed, the evidence from this study of photoperiodic changes the size of the nucleus of the FS cells would be consistent with direct effects of melatonin on this cell type. So, on this basis the authors should not only make reference to the changes in the FS in the results, but also add a sentence to the discussion to leave open the contribution of the FS cells, which as yet is unclear.

Reviewer #2 (Remarks to the Author):

I found that the paper itself is a very important contribution to the study of photoperiodic control of reproduction. I am largely satisfied with the authors' responses to my and the other reviewers' comments. While I recognize limitations is space and I know other authors have discussed the issues, I still feel the authors could have mentioned aspects concerning differences and similarities of these short day breeders with long day breeders.

Reviewer #3 (Remarks to the Author):

I am satisfied with the corrections made by the authors to my comments.

Reviewer #4 (Remarks to the Author):

I am happy with the response to my suggestions, except that I couldn't find a statistical comparison between the frequency distributions of Pearson correlation coefficients of seasonal versus non-seasonal genes (Figure 1d).

Response to reviewers: NCOMMS-20-01753A

Author responses in red.

Reviewer #1 (Remarks to the Author):

The authors have addressed the comments in full and have provided some comprehensive responses. In general terms I am content with the responses provided, but there are couple of minor issues for consideration.

Point 1: It is accepted that as antibodies are not currently available against Bmal2 and therefore the authors are currently unable to resolve the composition of the transcriptional complex. In the light of this the additional statement added to the text provides helpful caution.

We have added the following statement to the manuscript:

Page 8 line 195-199: "This suggests that BMAL2 operates as a co-activator of EYA3, in a CLOCK/BMAL1-dependent manner, requiring PAS-B dependent protein-protein interactions for the mechanism of action, confirmation of this will require suitable antibodies to be raised for future studies."

Point 2: Error corrected

Thank you for pointing this out to us.

Point 3: The explanation offered is helpful and accepted. In particular, the authors agree that there is desynchrony in the second peaks in BMAL2 and EYA3 and they offer a logical explanation why this is unlikely to be mechanistically important—namely the second peak in EYA3 is not observed early following switch to LP. However, the authors should also make this clear in the MS, by adding a sentence of explanation to this effect in the discussion.

We make reference to this second peak on page 9 line 240 of the manuscript. We have added the following statement to the discussion, page 12 line 313-317: "We note that both BMAL2 and EYA3 show a second peak on day 28 of LP, however, these peaks are "dysynchronised" (ZT20 BMAL2 and ZT15/16 EYA3). Furthermore, the second peak in EYA3 expression is absent in early responses to LP indicating that it is potentially not mechanistically important in a coincidence model". We feel further discussion of this will interrupt the flow of the manuscript. We will be making our response to review available publically and feel that the information will therefore be accessible to those readers that wish to unpick this further.

Point 4:

It is accepted that what drives Bmal2 expression is an important issue for future work and is beyond the scope of the current study.

We are glad the reviewer agrees.

Point 5: The point here is that the authors show data in supplementary Fig 1F, which shows clear photoperiodic changes in the size of the nucleus of the FS cells as well as the PT thyrotrophs. There is no comment or reference to these changes in the FS cells in the results section of the paper, but there is a clear statement that there are no morphological changes in the pars distalis somatotrophs. The authors use this contrast between the PT thyrotrophs and PD somatotrophs to support their

argument for increased chromatin accessibility which may be the basis for the photoinductive effects of LP leading to changes in the seasonal transcriptome.

While this is reasonable, they should nonetheless refer to the changes in the nuclear size of the FS cells in the results, and not just ignore them. In their rebuttal letter, the authors acknowledge that they cannot exclude the importance of the FS cells, but suggest they are less relevant to the current study on the basis that the PT thyrotrophs contain the melatonin receptors. Therefore, they focus their attention on the thyrotrophs as it receives the photoperiodic signal. While I agree that the PT thyrotrophs are the main output cell, there is no evidence (I know of) that the FS cells do not also express melatonin receptors. Indeed, the evidence from this study of photoperiodic changes the size of the nucleus of the FS cells would be consistent with direct effects of melatonin on this cell type. So, on this basis the authors should not only make reference to the changes in the FS in the results, but also add a sentence to the discussion to leave open the contribution of the FS cells, which as yet is unclear.

There is good evidence in rats that the PT thyrotroph is the only melatonin receptor expressing cell in the pars tuberalis (Dardenete 2002 <https://doi.org/10.1177/002215540205001209>) and the european hamster (Dardente 2002 <https://doi.org/10.1046/j.1365-2826.2003.01060.x>). Our own unpublished data seems to support this in sheep. However we take the reviewers point and have updated the manuscript accordingly:

Page 5 line 99-101: “ These morphological changes were not seen in *Pars distalis* (PD) somatotrophs but were observed in the PT follicular stellate (FS) cells to a lesser degree (Fig. 1c, Supplementary Fig. 1e & f)”.

Page 13 line 365-367: “Our data do indicate morphological changes in another PT cell type, the FS cell, however these cells lack melatonin receptors^{59,60} therefore a role in a coincidence timer is unclear.”

Reviewer #2 (Remarks to the Author):

I found that the paper itself is a very important contribution to the study of photoperiodic control of reproduction. I am largely satisfied with the authors' responses to my and the other reviewers' comments. While I recognize limitations in space and I know other authors have discussed the issues, I still feel the authors could have mentioned aspects concerning differences and similarities of these short day breeders with long day breeders.

We have added the following statement to the manuscript:

Page 14 line 372-375: “Even amongst long day and short day breeders the EYA3-TSH circuitry behaves similarly, demonstrating that although the downstream reproductive responses to photoperiod are altered the mechanism of photoperiodic time measurement is shared^{9,13”}”.

Reviewer #3 (Remarks to the Author):

I am satisfied with the corrections made by the authors to my comments.
Thank you for your helpful comments.

Reviewer #4 (Remarks to the Author):

I am happy with the response to my suggestions, except that I couldn't find a statistical comparison between the frequency distributions of Pearson correlation coefficients of seasonal versus non-seasonal genes (Figure 1d).

We apologise and have added this information to the manuscript and here for reference: P value < 0.001 (3.96e-25) by Mann-Whitney U test. The distributions of non-seasonal correlation and seasonal correlation of fig 1d were significantly different.

The manuscript has been updated:

Page 6 line 120-123: "Importantly, this correlation was absent in non-seasonally regulated genes and we found the distributions between seasonal and non-seasonally regulated genes to be significantly different (Fig. 1d, P value < 0.001 Mann-Whitney U test)".